# A one-two punch targeting reactive oxygen species and fibril for rescuing Alzheimer's disease

Jiefei Wang[1,7], Ping Shangguan[1,7], Xiaoyu Chen[2], Yong Zhong ®[3], Ming Lin[1], Mu He[1], Yisheng Liu[1], Yuan Zhou[1], Xiaobin Pang[1], Lulu Han[1], Mengya Lu[1], Xiao Wang[1], Yang Liu ®[1], Huiqing Yang[1], Jingyun Chen[1], Chenhui Song[1], Jing Zhang ®[4] ✉, Xin Wang ®[1] ✉, Bingyang Shi ®[1,5] ✉ & Ben Zhong Tang ®[6] ✉

Toxic amyloid-beta (Aβ) plaque and harmful inflammation are two leading symptoms of Alzheimer's disease (AD). However, precise AD therapy is unrealizable due to the lack of dual-targeting therapy function, poor BBB penetration, and low imaging sensitivity. Here, we design a near-infrared-II aggregation-induced emission (AIE) nanotheranostic for precise AD therapy. The anti-quenching emission at 1350 nm accurately monitors the in vivo BBB penetration and specifically binding of nanotheranostic with plaques. Triggered by reactive oxygen species (ROS), two encapsulated therapeutic-type AIE molecules are controllably released to activate a self-enhanced therapy program. One specifically inhibits the Aβ fibrils formation, degrades Aβ fibrils, and prevents the reaggregation via multi-competitive interactions that are verified by computational analysis, which further alleviates the inflammation. Another effectively scavenges ROS and inflammation to remodel the cerebral redox balance and enhances the therapy effect, together reversing the neurotoxicity and achieving effective behavioral and cognitive improvements in the female AD mice model.

As the population ages, the central nervous system (CNS) neurodegenerative disease prevalence gradually grows, posing a significant public health crisis[1,2]. Alzheimer's disease (AD) is one of the most common neurodegenerative diseases, impairing brain health and leading to severe neurological dysfunction and dyskinesia, which has been considered an incurable disease[3–5]. Toxic amyloid-beta (Aβ) plaque and harmful inflammation are the leading hallmarks of AD, reinforcing each other and aggravating the devastating process, i.e., Aβ deposits activate inflammation[6–8]. In return, the inflammation further exacerbates the deposition of Aβ plaque[9], together inducing the downstream free radical accumulation to mediate irreversible neuronal demise and synapse elimination[10]. Therefore, there is an urgent need to design an Aβ and inflammation dual-targeted therapeutic system against AD. Over the past decades, only several classes of

[1]Henan-Macquarie Uni Joint Centre for Biomedical Innovation, Academy for Advanced Interdisciplinary Studies, Henan Key Laboratory of Brain Targeted Bio-nanomedicine, School of Life Sciences, Henan University, Kaifeng, Henan 475004, China. [2]School of Medical Technology, Beijing Institute of Technology, Beijing, China. [3]Key Laboratory for Special Functional Materials of Ministry of Education, National & Local Joint Engineering Research Center for High-efficiency Display and Lighting Technology, School of Materials Science and Engineering, Collaborative Innovation Center of Nano Functional Materials and Applications, Henan University, Kaifeng 475004, China. [4]Department of Laboratory Medicine, Nanfang Hospital, Southern Medical University, Guangzhou 510515, China. [5]Macquarie Medical School, Faculty of Medicine & Health Sciences, Macquarie University, Sydney, NSW 2109, Australia. [6]School of Science and Engineering, Shenzhen Institute of Aggregate Science and Technology, The Chinese University of Hong Kong, Shenzhen, Guangdong 518172, China. [7]These authors contributed equally: Jiefei Wang, Ping Shangguan. ✉e-mail: zhangjingzisefeng@163.com; wx@henu.edu.cn; bingyang.shi@mq.edu.au; tangbenz@cuhk.edu.cn

approved drugs were developed to treat AD although pouring tremendous investments, e.g., acetylcholinesterase inhibitors, N-methyl D-aspartate (NMDA) receptor antagonists, anti-inflammatory agents, and monoclonal antibody[11–13]. However, these drugs are mono-targeted, have low specificity to both Aβ and inflammation symptoms, and some exist high toxic risks, e.g., up to 40% of clinical AD trials received a high dose (10 mg kg⁻¹) of aducanumab appeared severe brain edema and hemorrhage[14]. What's worse, the blood-brain barrier (BBB) blocks most drugs into the brain[15,16], thus inducing a poor therapy outcome[12,17]. The development of precise and efficient drugs is dependent on visual in vivo research means. Therefore, it has an urgent demand to develop a visual therapeutic drug with BBB-penetrating, high-specificity, and dual-targeting potential to achieve safe, effective, and precise therapy for AD via high-sensitivity in vivo visual monitoring of BBB penetration and binding with Aβ plaque, reversing the Aβ plaques, and in situ anti-inflammatory.

High-sensitivity imaging tracking technology could guide the directional and precise design of drugs via the feedback of in vivo monitoring results. Among numerous imaging technologies, fluorescence imaging with non-radioactive, noninvasive, safe, and real-time observation merits[18,19], is an ideal candidate to fabricate visual drugs for AD. Although a growing number of in vitro detected probes of Aβ plaques have already been developed, e.g., thioflavin probes (ThT or ThS)[20,21], unfortunately, these reported probes only function as a reporter for ex vivo histological staining of amyloid fibrils and are unable to support in vivo imaging and therapeutic applications for the following reasons: First, the difficulties in penetrating BBB and short half-life together block the penetration of probes into the brain. Secondly, all current probes only possess short emission wavelengths below 715 nm even those so-called NIR probes[22–25], far from the high-sensitivity NIR-II region (1000-1700 nm). In this case, these probes only acquired low imaging sensitivity that is proportional to wavelength. Moreover, the aggregation-caused quenching (ACQ) effect[26,27] and poor Aβ-affinity[28] further impaired the imaging sensitivity. Most importantly, all current probes lack active centers for specific dual-targeting therapy. These above requirements are way out of the league of current probe or drug molecule systems.

Therapeutic-type NIR-II probes with aggregation-induced emission (AIE) property, i.e., visual drugs, are supposed to hold noninvasive, large penetration depth[29,30], anti-quenching[31–33], and high signal-to-noise ratio (S/N) merits[34,35], as well as potential drug activity. As an advanced conceptual visual drug, it was expected to not only support the high-fidelity visualization of the BBB-traversing process and binding process of drugs with Aβ plaques but also offer a chance to achieve the above dual-targeted therapy for AD. However, there exist huge challenges to acquiring this ideal probe: (i) the modest hydrophobic skeleton with strong Aβ-affinity is the basis of Aβ fibril-targeted therapy-type probe[36–38]. (ii) large π-conjugated structures with intermolecular π-π stacking interaction can achieve long-wavelength absorption (spectral broadening) but the corresponding quantum yield ($\Phi_{PL}$) of emission was low, whereas twisted structures including some special aromatic rotors and bridged groups can acquire relatively high $\Phi_{PL}$ but generally result in low molar extinction coefficient ($\varepsilon$) value[27,39,40], thereby a balanced structure is needed for the highly bright, anti-quenching, and long-wavelength NIR-II emission. (iii) the multi-interacted sites and strong Aβ-affinity were necessary for the probe, which could pull back the Aβ monomer from the strong aggregated interaction with another Aβ monomer via the competition interaction. The site number and strength of competition determine the efficiency of the inhibition to fibril generation and fibril degradation. (iv) a safe and effective anti-inflammatory active center is another critical factor for therapy activity. (v) suitable delivery, release, and targeted modification forms need to be considered to acquire long in vivo circulation, superior BBB penetration, and maximize the drug concentration in AD lesion regions. To date, there is still no successful

paradigm that can address the above challenges to achieve such an ideal visual drug with superior NIR-II emission due to the lack of proper design strategies.

In this work, we fabricate a brain-targeting, fibril-degrading, and ROS-regulating NIR-II nanotheranostic system via ingenious molecule design and co-assembly strategy to successfully resolve the above long-standing challenges in AD. Firstly, we synthesize two theranostic-type AIE luminogens (AIEgens) with NIR-II emission, i.e., compound **3** (green block, cLogP = 7.63) and compound **6** (blue block) through imaginative multi-optimization and metal complex strategies. Then, they are further fabricated into reactive oxygen species (ROS)-responsive nanotheranostic via smart co-assembly engineering strategy and brain-targeted angiopep-2 (Ang-2) modification[41]. As shown in Fig. 1, the resulting nanocomposites (Ang-AIE NCs, termed Ang-NCs) generate new advantages neither existing in two AIEgens: (i) two hydrophobic molecules are changed to hydrophilic NCs with long in vivo elimination half-life (3.9 h), which is favorable for in vivo long-term tracking and the improvement of therapy efficacy. (ii) the NCs possess the longest emission at 1350 nm in the AD field to date, which generates the highest sensitivity for efficiently monitoring the across-skull-signals of in vivo Aβ plaque binding with NCs after traversing BBB. (iii) the inflammation-associated reactive oxygen species in the AD region triggers the controllable release of two AIEgens from NCs, generating an additively and synergistically self-enhanced therapy effect. The released compound **3** specifically inhibits the Aβ fibrils formation and disassembles the Aβ plaques via Van der Waals forces, hydrogen bonding, and π-π interactions. The compound **6** scavenges the harmful inflammation-associated ROS by the Ce(III) active center and alleviates the formation of Aβ fibrils. Moreover, the NCs strongly inhibit the potent harmful reaggregation process of the degraded Aβ fragments, remodel the cerebral redox balance, and reverse neurotoxicity to acquire remarkable behavioral and cognitive improvements in the AD mice model. Overall, the achievement of the NIR-II dual-targeting strategy, as a proof-of-concept, has well settled the dilemma of the Aβ fibril and inflammation in AD, thus might inspire more advanced visual nanotheranostic for other neurodegenerative diseases to guard our brain health.

## Results

### Rational design and characterization of compounds 3 and 6

Based on the above guidelines, we synthesized two therapeutic-type NIR-II AIEgens, i.e., compounds **3** and **6**, according to the route as shown in Supplementary Figs. 1–4. The intermediates have been confirmed by nuclear magnetic resonance (NMR) and high-resolution mass spectrum (HRMS) (Supplementary Figs. 5–14). Two AIEgens contain the same central backbone consisting of a benzo[1,2-c:4,5-c′]bis([1,2,5]thiadiazole (BBTD) acceptor unit, side alkyl chain-substituted thiophene bridging units and the same aromatic bridge groups (Fig. 2a). Differently, two symmetrical ends of **3** are 2,2′-azanediylbis(ethan-1-ol) groups that could specially bind with Aβ fibrils[42,43]. The ends of **6** are terpyridyl coordinated Ce(III) units, which endow **6** with effective antioxidation capability[44–46]. Actually, **6** is a polymer according to the dual coordination mode of Ce(III) centers, but not a small molecule.

According to the frontier molecular orbitals (FMOs), the results indicated the electron cloud of the lowest unoccupied molecular orbital (LUMO) of **3** in ground-state (S₀) is dominantly localized in the core, while the highest occupied molecular orbital (HOMO) is delocalized across the whole conjugated backbone (Fig. 2b). The corresponding energy gap between HOMO and LUMO was calculated to be 1.40 eV, revealing an apparent donor-acceptor (D-A) interaction and intramolecular charge transfer effect from D/π-bridge to BBTD core contributing to NIR-II absorption and emission[47]. By contrast, the quenching effect of central Ce(III), **3** has intense NIR-II emission. The main contribution may be generated by the two dihedral angles (46.3°

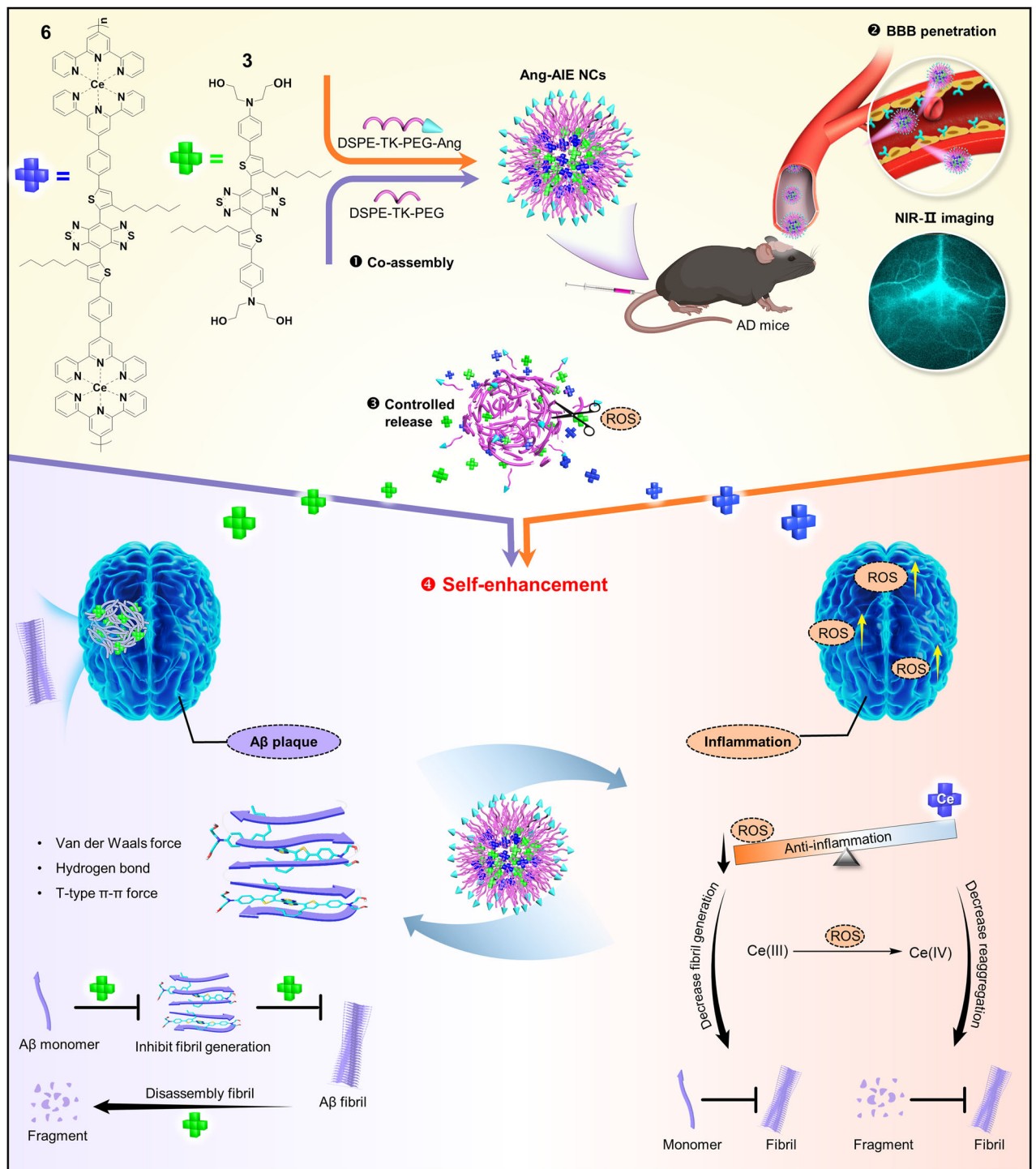

**Fig. 1 | Schematic of NIR-II brain-target theranostic system for dual-target therapy of AD via a four-step route.** (i) The co-assembly of nanodrug (Ang-AIE NCs, termed Ang-NCs). (ii) the NIR-II imaging monitored the BBB-traversing process of Ang-NCs. (iii) the inflammation-associated reactive oxygen species (ROS) environment stimulated the controllable release of Ang-NCs. iv) the activation of the self-enhanced dual-targeting program: compound **3** specifically inhibits the formation of amyloid-beta (Aβ) fibrils, degrades toxic fibrils, and inhibits reaggregation. Compound **6** relieves harmful inflammation-associated ROS and promotes the inhibition of Aβ fibrils generation.

and 46.6˚) between the BBTD core and adjacent thiophene group of optimized $S_0$ state (Supplementary Fig. 15), and 48.3˚ and 88.3˚ of the excited state ($S_1$) accompanied by 0.83 eV gap (Supplementary Fig. 16).

**Fabrication and optical performance characterization of NCs**

We first verified the emission character of two AIEgens. The results revealed typical aggregation-induced photoluminescence (PL)

increase, suggesting both were typical AIE molecules (Fig. 2c, Supplementary Fig. 17). To achieve the long in vivo circulation, hydrophilicity, and BBB penetration, two AIEgens were co-assembled into nanocomposites via a ROS-responsive thioketal (TK)-containing template (DSPE-TK-PEG)[48] combined with angiopep-2 (Ang-2) modification. The obtained Ang-NCs revealed amorphous morphology (inset) with an average size centered at 153.4 nm (Fig. 2d). As expected, the

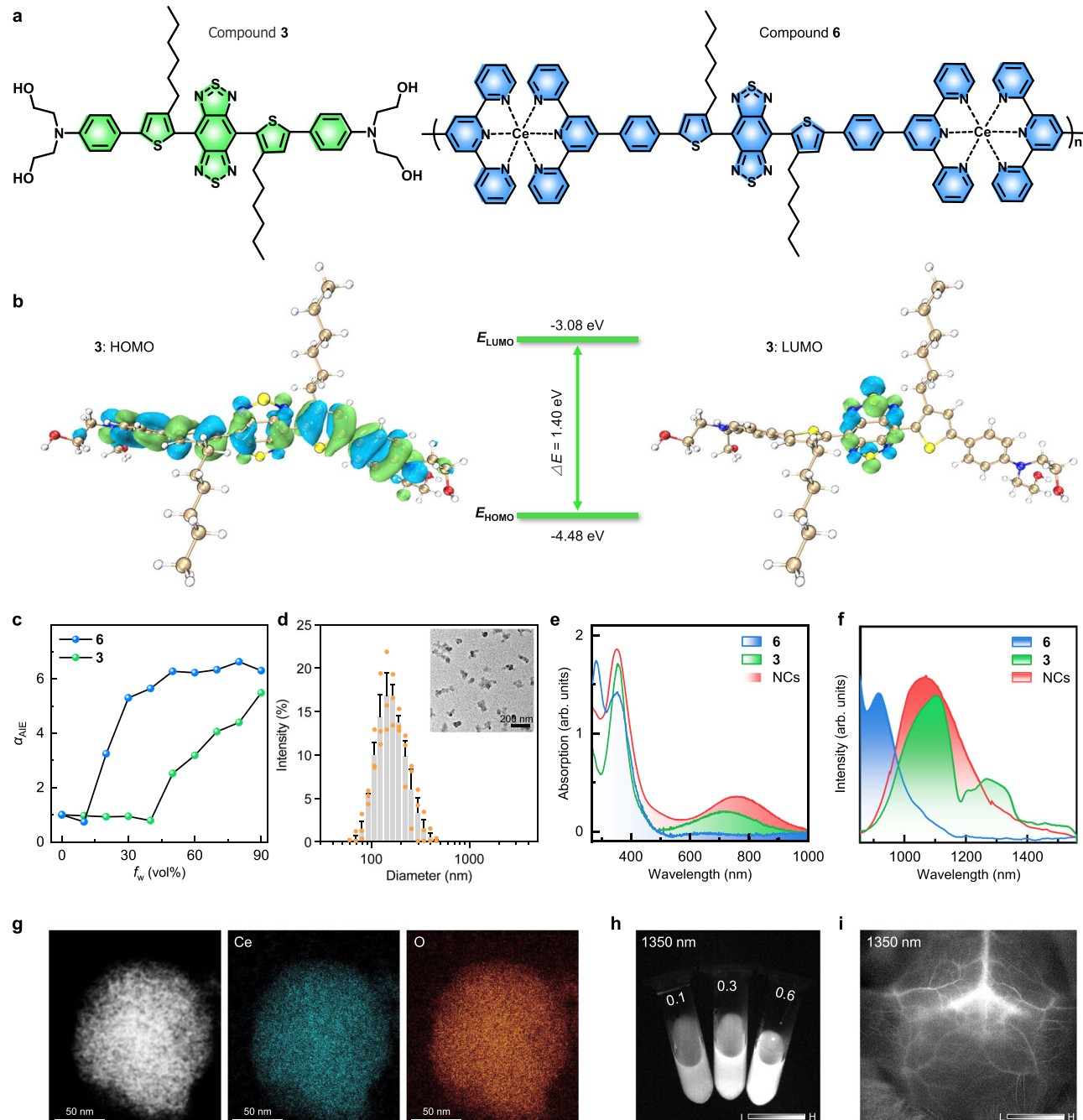

**Fig. 2 | Photophysical property characterization of two AIE luminogens (AIEgens) and NCs. a** Chemical formula of **3** and **6**. **b** Illustration of the HOMO/LUMO and the corresponding energy gap of **3** in the ground-state (S$_0$). **c** Increase ratio ($\alpha_{AIE}$) of relative photoluminescence (PL) intensity of **3** at 1110 nm and **6** at 916 nm. **d** Representative size distribution and morphology image of NCs. The data was presented as mean ± SEM. **e**, **f** were respectively the absorption and emission spectra (excitation: 808 nm) of two AIEgens and NCs. **g** Elements (Ce and O) mapping of NCs. **h** NIR imaging at 1350 nm for various concentrations of NCs aqueous solution (0.1, 0.3, and 0.6 mg/mL). **i** In vivo NIR imaging at 1350 nm of cerebral vessels in mice without removing the scalp or skull after *i.v.* injection of Ang-NCs. Source data are provided as a Source Data file.

absorption and maximum emission of NCs were respectively loaded at 760 nm and 1070 nm with tail emission exceeding 1550 nm as shown in Fig. 2e, f. The NCs mainly inherit the photonic performance of **3**. Two AIEgens were homogeneously distributed in the NCs (Fig. 2g). Fourier transform infrared (FTIR) spectra verified the Ang-2 modification (Supplementary Fig. 18). Visual NIR-II images of NCs solution indicated a concentration-/wavelength-dependent brightness (Fig. 2h, Supplementary Fig. 19). The NIR imaging at 1350 nm of Ang-NCs clearly outlines the frame of cerebrovascular (Fig. 2i) with higher sensitivity than that of other channels (Supplementary Fig. 20). Therefore, the window

at 1350 nm was selected than 1150 nm and 1550 nm due to the maximization of penetration depth and brightness.

## Antioxidation, fibril disassembly, and inhibition of fibril generation

We further evidenced the dual-target therapy function of NCs. For the anti-inflammation capacity, hydrogen peroxide (H$_2$O$_2$) was used to mimic the high-inflammation environment of the AD region characterized by high levels of ROS and toxic free radical (·OH). As shown in Fig. 3a, the controlled release of NCs was monitored by dynamic light

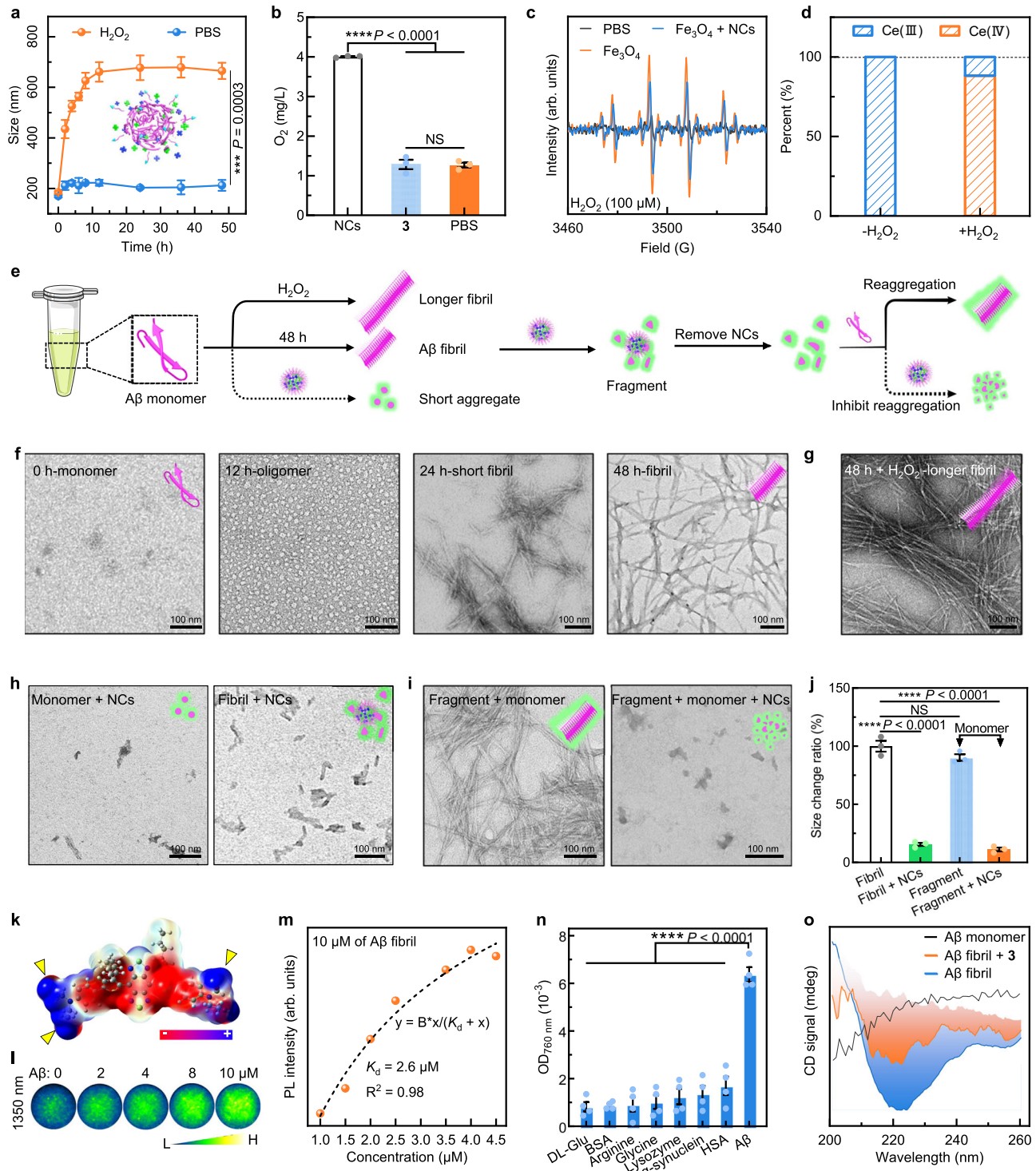

scattering (DLS) analysis, and the result revealed a gradually increased size with swelling and broken morphology (Supplementary Fig. 21), suggesting effective ROS-responsive release. The Ce(III) in the released **6** could catalyze $H_2O_2$ to oxygen (Fig. 3b) and significantly decreased the radical generated by iron oxide ($Fe_3O_4$) nanoparticles with peroxidase (POD)-like activity[49] (Fig. 3c). The Ce(III) was oxidized into Ce(IV) product by $H_2O_2$, which was evidenced by X-ray photoelectron spectroscopy (Fig. 3d, Supplementary Fig. 22), and revealed a potential antioxidation-mediated anti-inflammation capacity of our NCs.

Then we investigated the parallel therapy functions, i.e., inhibition of fibril generation, Aβ fibril disassembly, and reaggregation inhibition by NCs (Fig. 3e). We compared the morphology and length of various

Aβ species with different treatments by transmission electron microscope (TEM) observation and DLS determination. The Aβ monomer underwent an intermediated product, i.e., oligomer (12 h), to Aβ fibril after 48 h of growth (Fig. 3f). Moreover, we discovered that $H_2O_2$ could promote the elongation of Aβ fibrils according to the TEM image (Fig. 3g). As expected, our NCs not only inhibited the transform of monomer into fibril but also degraded Aβ fibril into amorphous fragments (Fig. 3h). In sharp contrast to this, the commercial Thioflavin T (ThT), Thioflavin S (ThS), and Pittsburg-B (PiB) probes were unable to inhibit the fibril generation and degrade Aβ fibril (Supplementary Fig. 23). The TEM images and corresponding size monitoring indicated an obvious restructuring from the broken fibril back into longer

**Fig. 3 | Anti-oxidation, selective inhibition of the formation for Aβ fibrils, Aβ fibril degradation, and reaggregation inhibition by NCs. a** Size change of NCs treated with PBS and hydrogen peroxide ($H_2O_2$) for various times. **b** Oxygen generation comparison of $H_2O_2$ that respectively was treated with PBS, **3**, and NCs. **a**, **b** were representative data of three independent experiments. **c** Electron spin resonance (ESR) spectra comparison among the PBS group, iron oxide nanoparticles ($Fe_3O_4$, 100 μg/mL), and the mixture of $Fe_3O_4$ with NCs, all contain $H_2O_2$. **d** Proportion change of various valence states for Ce element in NCs before and after $H_2O_2$ treatment. **e** Illustration for the experiment design, including the oxidation promoted the elongation of Aβ fibrils, NCs achieved the fibril degradation, and NCs inhibited the aggregation of monomer and reaggregation of fragments. **f** Transmission electron microscope (TEM) images of the morphology change of Aβ species from monomer, oligomer to Aβ fibrils during 48 h of growth. **g** TEM image of Aβ fibril treated with $H_2O_2$. **h** TEM images of the fibril degradation and the inhibition of fibril generation treated by NCs for 48 h. **i** TEM images for the reaggregation of fragments and reaggregation inhibition by NCs. **j** The corresponding length of fibrils tested by DLS under different conditions. **f**–**j** were representative data of three independent experiments. **k** Positive (blue) and negative (red) electrostatic potential analysis of the DFT-optimized **3**. **l** NIR imaging at 1350 nm of various concentrations of Aβ fibril with **3** (2 μM). **m** The monitoring of fluorescence intensity at 1060 nm of Aβ fibrils (10 μM) treated with various concentrations of **3**. **n** Absorption at 760 nm of various proteins, bovine serum albumin (BSA), and human serum albumin (HSA) treated with **3**. The absorption test was representative of three independent experiments. **o** CD spectra of Aβ monomer, Aβ fibril (5 μM), and Aβ fibril (5 μM) incubating with **3**. Unless otherwise stated, all Aβ peptides in whole assays were $Aβ_{42}$ (10 μM), and all concentrations of $H_2O_2$ and NCs were 100 μM and 100 μg/mL. All data were presented as mean ± SEM. NS, no significance, $^{***}P < 0.001$, and $^{****}P < 0.0001$. All statistical analyses were compared by analyses of one-way ANOVA except the two-tailed Student's *t*-test in **a**. Source data are provided as a Source Data file.

aggregated fibrils after growing for 48 h and replenishing the Aβ monomer, while it could be greatly inhibited by the NCs (Fig. 3i, j).

To verify the binding process, we first conducted the electrostatic potential analysis of the density functional theory (DFT)-optimized **3** (Fig. 3k). The result indicated that **3** has an obvious polarization with central negative conjugated core (repulsive interactions) and positive charge ends (attractive interactions), which showed huge potential to bind with negative Aβ species (Supplementary Fig. 24) via weak interactions. Furthermore, we used NIR fluorescence at 1350 nm to monitor the concentration-dependent binding either forward (Fig. 3l) or reverse titration (Supplementary Fig. 25). The dissociation constant ($K_d$) of **3** was determined from the reverse titration nonlinear fitting curve to be 2.6 μM (Fig. 3m), which is close to that of ThT ($K_d = 0.89$ μM)[50], responding to comparable binding affinity. Moreover, the ThT experiment on the inhibition kinetics of NCs against Aβ monomers was tested. The results indicated the 50% inhibiting concentration ($IC_{50}$) value of NCs is 9.67 μg/mL, suggesting effective inhibition of NCs against the aggregation of Aβ monomer (Supplementary Fig. 26). Notably, **3** also has a 7.7-fold binding selectivity with Aβ fibril than that of DL-Glutamine (DL-Glu, Fig. 3n), indicating a strong specificity with Aβ fibril. The bound **3** further disrupted the aggregated structure of Aβ fibril tested by the circular dichroism (CD) spectra (Fig. 3o).

## Aβ-degradation mechanism exploration via computation

To explore the detailed interaction of how **3** interferes with Aβ fibril. We attempted to use molecular dynamics (MD) simulations to find the detailed interaction force in the above Aβ fibril degradation, blocking the generation of fibril, and reaggregation inhibition process. We optimized the crystal structure of Aβ fibrils and performed geometry optimization on compound **3**. We obtained the crystal structure of $Aβ_{42}$ amyloid fibrils from the RCSB protein data bank (PDB code: 5KK3)[51] and prepared its structure by removing substructures, repairing sidechains, adding hydrogens and charges with AmberTools 20[52]. The DFT calculations were performed on **3** with the Gaussian 16 program package[53]. The geometry optimization of **3** was carried out at the B3LYP-D3 level of theory[54] with the 6-31 G(d) basis set. To predict how **3** interferes with the Aβ protein, we performed MD simulations on the protein in water for 100 ns, which allowed sufficient time for equilibration (Supplementary Fig. 27).

We utilized the protein structure obtained from 100 ns of MD simulations and docked it with the DFT-optimized structure of **3** using AutoDock Vina[55]. Next, we used the complex structure to perform MD simulations for 100 ns (Fig. 4a). General Amber force field (GAFF)[56] was used for **3**. The FF14SB[57] force field was used for the protein. The TIP3P force field was used for water molecules[58]. Finally, we analyzed the interactions between the Aβ protein and **3** using the results of MD simulations (Fig. 4b-d). The complex structure was shown at different snapshots with a time gap of 2 ns (Supplementary Fig. 28). According to the results of MD simulations, **3** (sticks) docked at the interface between two Aβ monomers (surface). We tallied the frequency of different residues observed in fifty plots to identify the active-site residues. We identified 16 active-site residues in the Aβ protein that interact with **3** at high frequencies (20-50), including H37, H100, G317, V318, V319, G349, V351, G381, V383, G413, V414, V415, G445, V447, V479 and V511 (Supplementary Table 1). The changes in distance between the mass center of **3** and 16 active-site residues of Aβ fibril during 100 ns MD simulations are listed in Supplementary Fig. 29. The detailed analysis revealed that Van der Waals force and hydrogen bonding (with V318 and V414 residues) between Aβ peptide with **3** were two of the main forces (Fig. 4b). Moreover, the partially negatively charged alkyl thiophene in **3** is stabilized by a T-shaped π-π stacking interaction with the aromatic rings of residues H100 and H37 (Fig. 4c).

The binding free energy was calculated by the molecular mechanics generalized Born surface area (MM/GBSA). We selected the equilibrium trajectory from the last 1 ns for calculation, which showed that MM/GBSA is -35.8 kcal/mol. From this calculation, we can see that **3** can bind to proteins more tightly. To accurately describe the binding affinity of **3** and protein, we performed high-precision quantum mechanical calculations. First, we used ONIOM (B3LYP-D3/6-31 G(d)//Amber) to determine the most stable structure. After conducting an ONIOM model calculation, we chose the structure with the lowest energy as the basis for our theozyme model. Meanwhile, we selected the structure of the final frame and computed the average distances from binding residues in the last 100 snapshots to **3**. Then we identified one with the strongest correlation between its distance and average distances obtained from MD simulations. We used the same DFT method for selected theozyme structure optimization. We calculated the binding free energy to be -8.5 kcal/mol, which is consistent with the experimental findings of -7.6 kcal/mol and further validates our previous analysis of the impact of **3** on protein binding forces. The data verified the function design of **3** as shown in Fig. 4e, the introduction of the hydrophobic central backbone, alkyl chain, 2,2'-azanediylbis(ethan-1-ol), S and O heteroatom for hydrogen bond, and conjugated thiophene and benzene ring (Supplementary Table 2).

To further verify if the contributed component was generated from **3** rather than **6** for Aβ fibril degradation, we detected the TEM images of Aβ fibril treated respectively with **3** and **6**. As expected, the fibril can be degraded by **3** but with negligible change by **6** (Fig. 4f). Collectively, the results corroborated that **6** and **3** in Ang-NCs respectively supported the antioxidation and Aβ-related therapy functions, i.e., inhibition of fibril generation and fibril degradation, which has incomparable advantages by all current commercial probes.

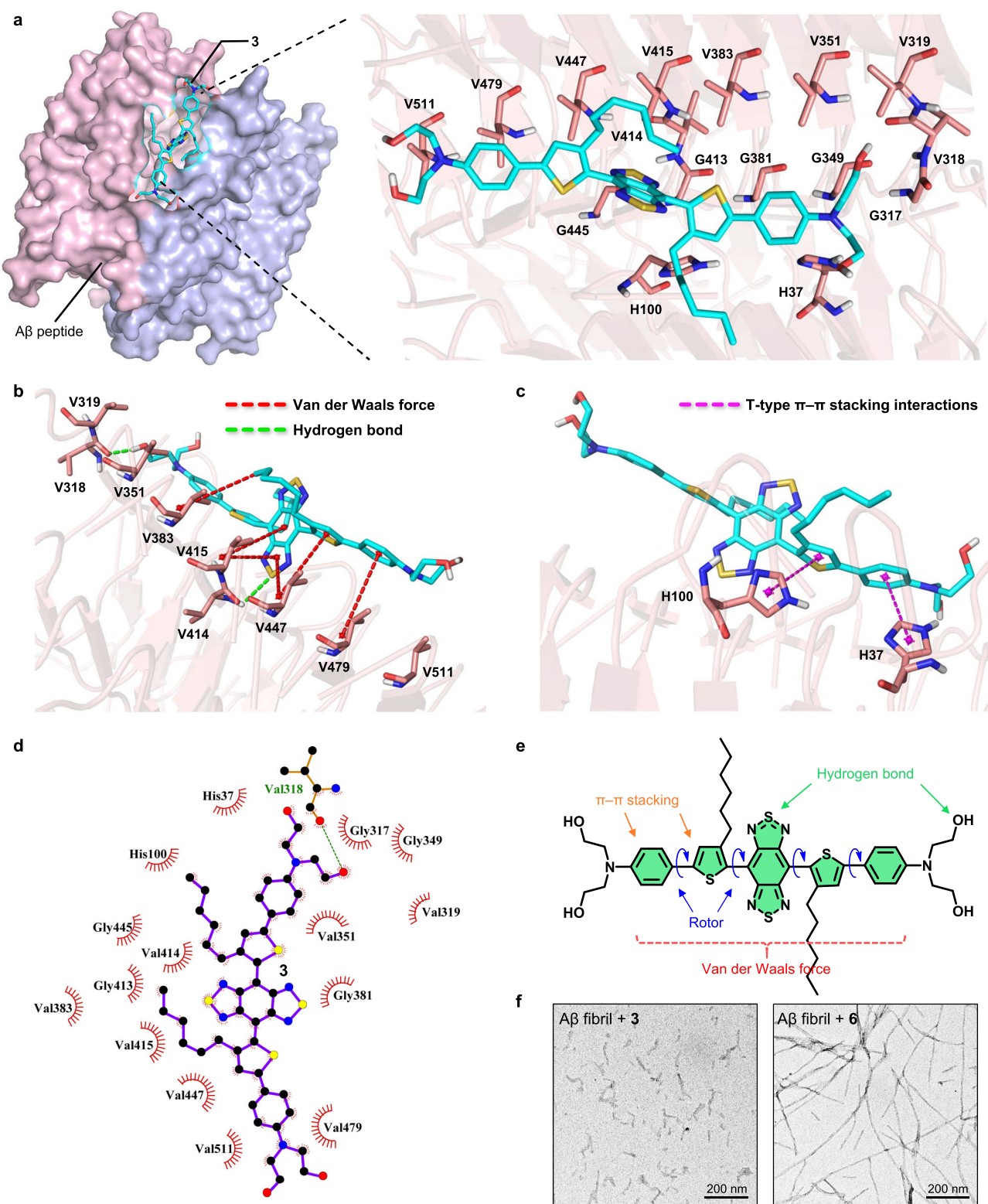

**Fig. 4 | MD results for the binding of 3 with Aβ protein. a 3** bound to the surface structure of Aβ$_{42}$ protein via 16 residues of 5KK3 that are close to **3**. **b** Van der Waals force and hydrogen bonds interactions between **3** and Aβ protein. **c** T-shaped π–π stacking interactions between **3** and Aβ protein. **d** Top view of the 16 active-site residues. **e** Binding groups analysis of **3**. **f** TEM images of fibril (25 μM) incubated with **3** and **6** molecules (25 μM). The TEM images were representative data from three independent experiments.

## In vitro synergistic therapy

Then, we evaluated the in vitro BBB permeability and synergistic therapy efficiency of Ang-NCs. First, we examined the cytotoxicity of the NCs and Ang-NCs. As illustrated in Fig. 5a, there have no significant toxicities were observed in both groups on various cell lines including SH-SY5Y, PC12, BV2, Neuro-2a, and HT-22 cells. The uptake of Ang-NCs by the PC12 cells was tested by the confocal images and quantified by the optical density in the 96-well plates, the results indicated that Ang-

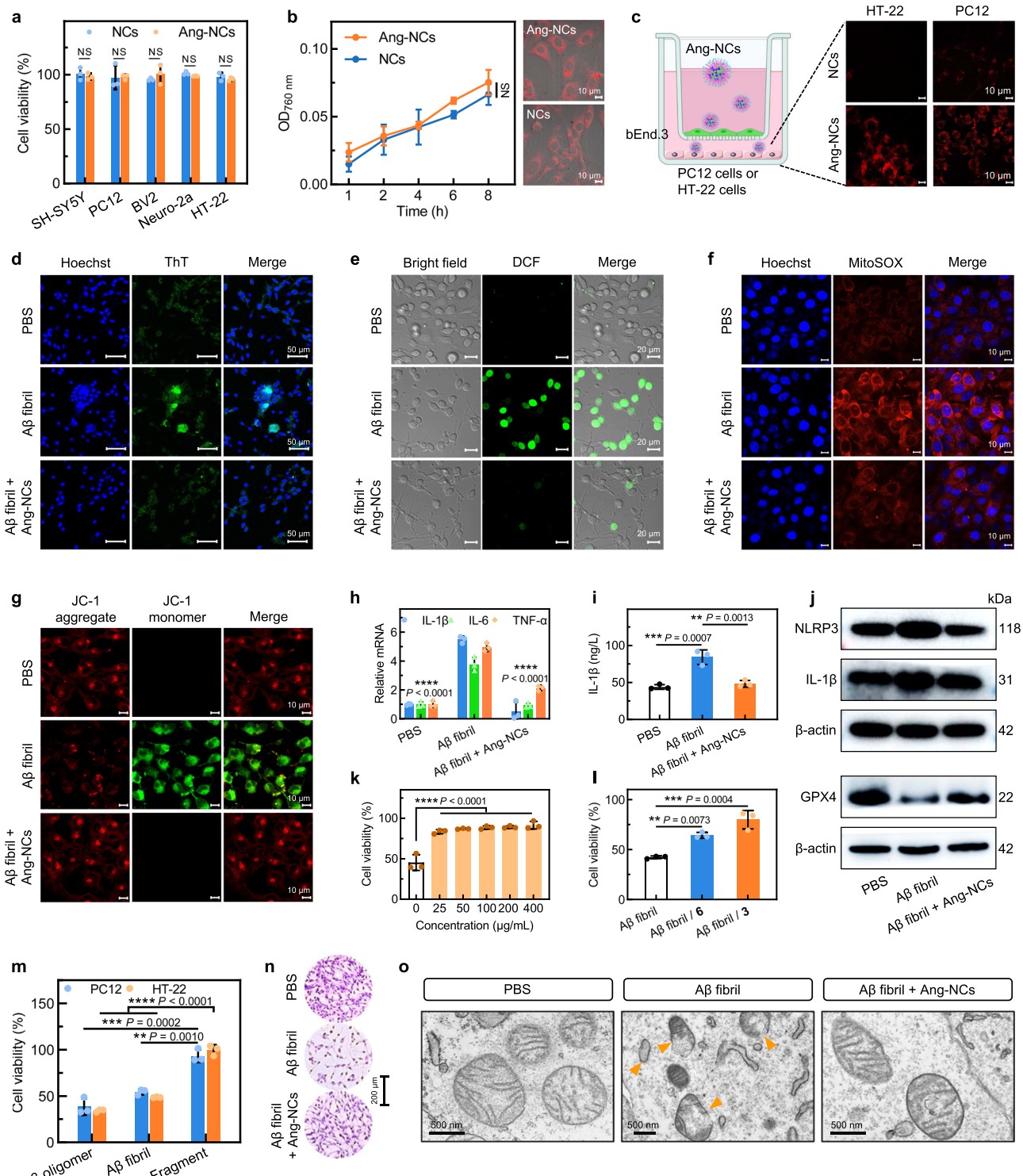

NCs have almost the same uptake as NCs (Fig. 5b). Furthermore, we respectively detected the in vitro BBB penetration of NCs and Ang-NCs in PC12 and HT-22 cells via Transwell model. Compared to the negligible fluorescence signals of NCs in two cells, the Ang-NCs displayed bright red fluorescence in two cells, implying enhanced BBB penetration by Ang-2 modification (Fig. 5c).

To assess the dual-target therapy ability for neuroprotection of Ang-NCs, we used confocal imaging observation to monitor the Aβ fibril degradation capability of Ang-NCs. The decreased green fluorescence from ThT strongly approved that Ang-NCs could efficiently degrade the Aβ fibril around the PC12 cells (Fig. 5d). For the antioxidant

function assessment of Ang-NCs, we detected the total ROS, mitochondria superoxide, and mitochondria damage respectively using 2'-7'dichlorofluorescin diacetate (DCFH-DA probe), Mitochondrial Superoxide Indicator (MitoSOX probe), and Mitochondrial Membrane Potential Kit (JC-1 probe). Compared to the Aβ fibril group, the Ang-NCs group revealed inhibited ROS level with weak green fluorescence from DCF (Fig. 5e), decreased red signal from MitoSOX (Fig. 5f), and suppressive green/red fluorescence ratio of JC-1 (Fig. 5g), which were the same as that of the PBS group, verify a strong antioxidant capability. In addition, all mRNA levels of pro-inflammatory factors (TNF-α, IL-1β, and IL-6) of the Ang-NCs group tested by RT-PCR assay showed

**Fig. 5 | In vitro neuroprotective effect of Ang-NCs. a** Cell viability of various cell lines treated with NCs and Ang-NCs ($n = 3$ biologically independent samples). **b** The confocal images and quantification in the 96-well plates for the uptake of NCs and Ang-NCs in PC12 cells ($n = 3$). **c** Confocal images of in vitro BBB penetration of two NCs by PC12 and HT-22 cells via Transwell model. **d** Confocal images of Aβ fibril degradation in PC12 cells with different treatments and ThT staining (green). **e** Cellular ROS in PC12 cells with various treatments for 48 h and stained with DCFH-DA (green). **f, g** Confocal images of the superoxide generation and mitochondrial membrane potential in HT-22 cells with various treatments for 48 h and staining by MitoSOX (**f**) and JC-1 probes (**g**). The confocal images were representative data of three independent experiments in **c**–**g**. **h** The relative expression of mRNA (TNF-α, IL-1β, and IL-6) in HT-22 cells tested by RT-PCR ($n = 3$). The $P$ values (***) were the comparison of three parameters between the Aβ fibril group and the other two groups, respectively. **i** IL-1β level in HT-22 cells tested by ELISA kit ($n = 3$). **j** The expression of IL-1β and NLRP3 in HT-22 cells, and GPX4 in PC12 cells by Western blot analyses. **k** Cell viability of PC12 cells treated with various concentrations of Ang-NCs ($n = 3$). **l** Cell viability of PC12 cells treated with Aβ fibril, Aβ fibril with **6** (100 μg/mL), and Aβ fibril with **3** (100 μg/mL) for 48 h ($n = 3$). **m** Cell viability ($n = 3$) of PC12 and HT-22 cells incubated with Aβ oligomer (10 μM), Aβ fibril, and degraded Aβ fragment (10 μM). **n** Representative crystal violet staining images of HT-22 cells after various treatments for 48 h. **o** The corresponding biological transmission electron microscopy (Bio-TEM) images of mitochondria in HT-22 cells with various treatments. Unless otherwise stated, all concentrations of Aβ fibril and Ang-NCs were respectively 10 μM and 100 μg/mL. All data were presented as mean ± SD. NS, no significance, **$P < 0.01$, ***$P < 0.001$, and ****$P < 0.0001$. All assays ($n = 3$) were biologically independent samples. All statistical analyses were compared by analyses of one-way ANOVA except the two-tailed Student's $t$-test in **a** and **b**. Source data are provided **a**s a Source Data file.

decreased levels, which were close to that of PBS group (Fig. 5h). The enzyme-linked immunosorbent assay (ELISA) result revealed an obvious content decrease of IL-1β in Ang-NCs with Aβ fibril group than Aβ fibril group (Fig. 5i). Then we further verified the recently identified AD-associated IL-1β/NLRP3 inflammatory pathway and antioxidant regulator glutathione peroxidase 4 (GPX4) by Western blot test. The downregulated IL-1β/NLRP3 and upregulated GPX4 in the Ang-NCs with Aβ fibril group indicated a strong antioxidation and anti-inflammation capability (Fig. 5j). The results suggested Aβ fibril-induced a severe inflammation response and Ang-NCs could reverse it through strong antioxidant capability, implying a potential neuroprotective effect.

Subsequently, the neuroprotective effect was explored in detail. First, the cell viability of various groups was detected and the result indicated the Ang-NCs could reverse the cell toxicity induced by Aβ fibril and inflammation, indicating a gradually increased cell viability as a function of concentration (Fig. 5k). To further verify the therapy contribution, we respectively detected the disassembly function of **3** and **6** against Aβ fibril and ROS-alleviating capability, the confocal imaging result indicated the efficient degradation of Aβ fibril was contributed by **3** rather than **6** (Supplementary Fig. 30a). On the other hand, the intensity quantification of DCF tested by a microplate reader revealed a strong antioxidation effect of NCs, suggesting **6** was the main contributor to the antioxidation (Supplementary Fig. 30b). As a result, both two molecules could reverse the toxicity of the Aβ fibril and fibril-induced oxidative damage (Fig. 5l). Then, we further evaluated the toxicity of degraded Aβ fragment to PC12 and HT-22 cells. The result indicated negligible toxicity of the Aβ fragment to two cells (Fig. 5m). Lastly, the neuroprotective effect was verified by crystalline violet imaging, in which the cell density visually represents the cell viability. The representative images powerfully verify that the Ang-NCs group with large cell density has a superior neuroprotective function (Fig. 5n). The Bio-TEM images visually detected the state of mitochondria. The HT-22 cells treated with Aβ fibril appeared to severe destruction of the mitochondria membrane and disappearance of cristae. By contrast, the Ang-NCs group could reverse the damaging effect and displayed intact morphology of mitochondria as that of PBS treatment (Fig. 5o). Overall, the data demonstrated our Ang-NCs have strong in vitro neuroprotection through scavenging excess ROS and inflammation, as well as fibril degradation.

## In vivo synergistic therapy effects on AD mice

Lastly, we investigated the in vivo BBB penetration and therapeutic function of Ang-NCs in AD mice according to the workflow (Fig. 6a), which includes the *i.v.* injection of PBS or Ang-NCs once every three days and performed six times of injections. First, it was commenced with a biocompatibility assessment. The hemolysis assay was investigated, and the colorless supernatant indicated a low hemolysis ratio and high safety of Ang-NCs treatment (Supplementary Fig. 31). The long half-life of the nanoparticle also was verified by the

pharmacokinetic test, and the result indicated a longer half-time (4.8 h) of the targeted group than 3.9 h in the non-targeted group (Fig. 6b), which was favorable for long-acting in vivo BBB penetration that was monitored by the NIR-II imaging at 1350 nm. As a result, the Ang-NCs group indicated much higher fluorescence intensity than that of the NCs group due to stronger BBB penetration and higher binding with Aβ fibril in the brain (Fig. 6c). Moreover, the NIR images and corresponding intensity quantification together indicated the major accumulation were distributed in the liver and spleen (Fig. 6d, e).

To explore the in vivo dual-targeted therapy function of Ang-NCs in the female AD mice after six times of injections, we first analyzed in situ ROS change in the ex vivo brain after *i.v.* injection of Ang-NCs one time, the results revealed an efficient decrease of excess ROS induced by Aβ fibril and inflammation (Fig. 6f). Then, reduced green fluorescence signals and intensity of ROS level in the brain slices stained by 8-hydroxy-2′-deoxyguanosine (8-OHdG)[59] verified the Ang-NCs could efficiently decrease the oxidation level in the brain of APP/PS1 mice (Supplementary Fig. 32). Moreover, the level change of TNF-α, IL-1β, and Aβ$_{1-42}$ in the mice indicated high in vivo anti-inflammatory capability and strong Aβ degradation capability of Ang-NCs (Fig. 6g). Subsequently, the Aβ plaques degradation and whether the existed activation of astrocytes and microglia cells were explored in detail. The ThS probe (Supplementary Fig. 33) and a specific Aβ probe (Rabbit pAb anti-Aβ) for Aβ plaques were used to detect the Aβ plaque in the hippocampus and cortex of ex vivo slices. A mass of plaques was observed in the AD mice. In contrast, the Ang-NCs powerfully degraded the plaque and revealed weak fluorescence as that of the wild-type (WT) group (Fig. 6h, i). The observation of microglia and astrocytes assisted by Iba-1 and GFAP staining showed decreased red fluorescence levels in the Ang-NCs groups than AD mice (Fig. 6i–k), suggesting no obvious activation of astrocytes and microglia cells[60]. Moreover, the cerebral microhemorrhage risk of Ang-NCs treatment was investigated by the Prussian Blue histochemistry of brain slices. The microhemorrhage profile images revealed the AD mice with Ang-NCs treatment have no obvious iron-positive deposits of microbleeds as to that WT mice, reflecting the high safety of Ang-NCs treatment (Supplementary Fig. 34).

Subsequently, we verified the improvement of the above dual-target therapy on memory loss, cognitive impairment, and behavioral impairment in AD mice post the treatment of Ang-NCs. Firstly, the nest construction (NC) test was performed to evaluate the improvement of motor deficits. Compared to the typical unimpaired paper pieces and the lowest behavior score in the AD group (Fig. 6l, m), the Ang-NCs group displayed a better performance and higher behavior score than that of the AD group. Subsequently, the long-term learning and memory capability in the spatial orientation of various groups were investigated by the Morris water maze (MWM) test. The time spent moving from the start site to the target platform (escape latency) was recorded in the first five days. After removing the platform, the moving route, crossing number, and staying time in the target quadrant were

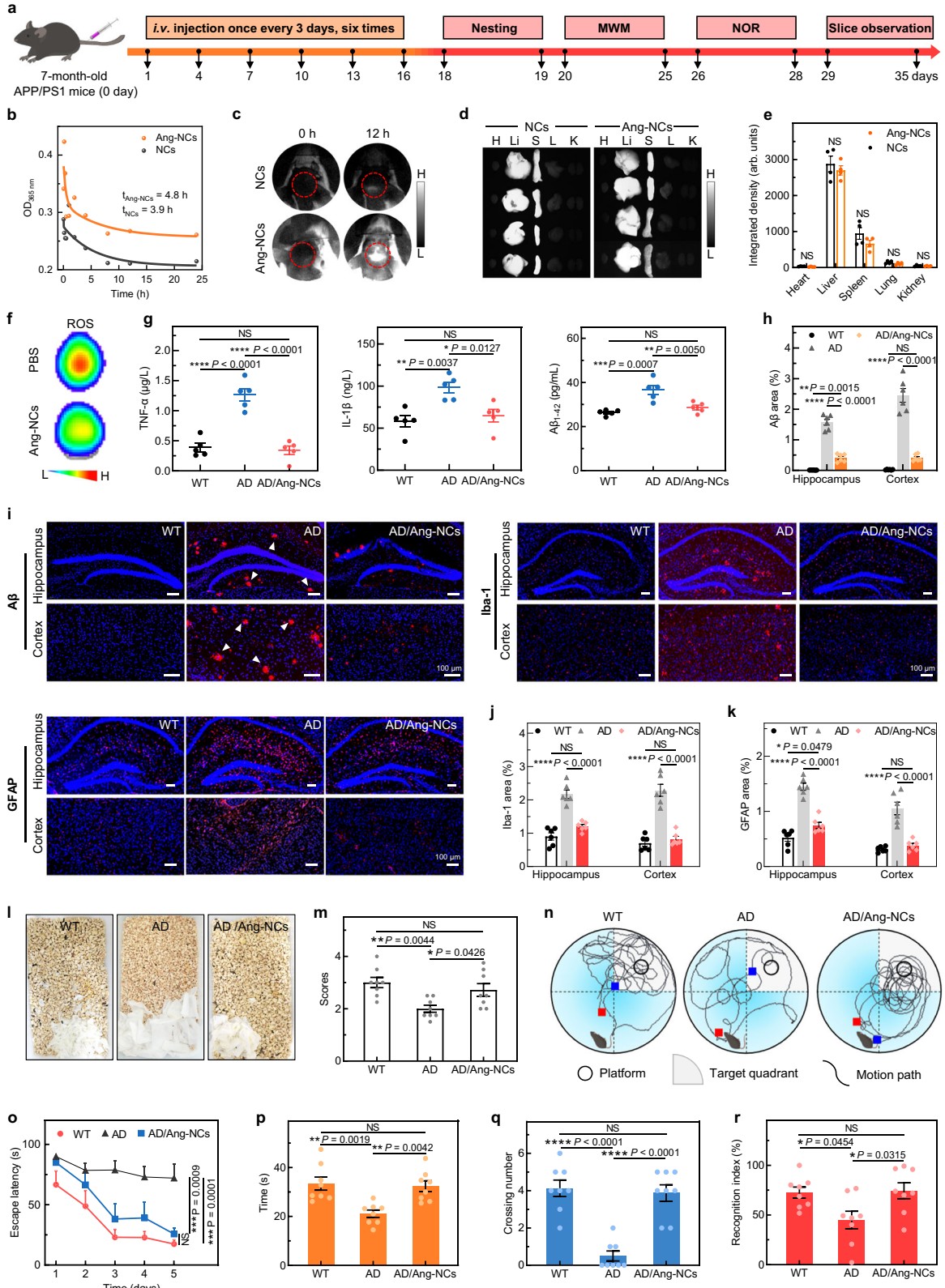

recorded on the sixth day. Both the WT and Ang-NCs groups revealed a purposeful route to the platform and gradually decreased escape latency compared to the AD group (Fig. 6n, o). Moreover, the residence time in the target quadrant (Fig. 6p) and crossing number through the platform (Fig. 6q) of the Ang-NCs group have significant improvements compared to that of the AD group. At last, the novel object recognition (NOR) test was further monitored. The Ang-NCs

group demonstrated a higher recognition index (RI) compared to that of AD mice (Fig. 6r), which was comparable to that of the WT mice, reflecting great interest in exploring novel objects. These results together demonstrated the Ang-NCs treatment could significantly improve the activity of daily living, spatial learning memory, and recognition capability of AD mice. We further examined the biosafety through the hematoxylin and eosin (H&E) staining of major organs

**Fig. 6 | In vivo theranostic of Ang-NCs in the female APP/PS1 (AD) mice. a** The outline of the experiment design and sequences for animal behavior evaluation. **b** In vivo pharmacokinetic analyses (*n* = 3 biologically independent mice). **c** NIR imaging at 1350 nm of the brain in APP/PS1 mice at various time points (0 and 12 h) post *i.v.* injection of NCs and Ang-NCs (dosage: 10 mg/kg). **d**, **e** NIR images and corresponding fluorescence quantification for main organs of AD mice treated with NCs and Ang-NCs (*n* = 4 biologically independent mice), heart (H), liver (Li), spleen (S), lung (L), and kidney (K). **f** Fluorescence images of ROS in ex vivo brains from AD mice post the injection of PBS and Ang-NCs. The DCFH-DA was dropped onto the surface of ex vivo brains before imaging. **g** The levels of TNF-α, IL-1β, and Aβ$_{1-42}$ in the various mice (*n* = 5 biologically independent samples). **h** Fluorescence quantification of Aβ plaques of brain slices (3 biologically independent mice and respectively sampling two tested sites in the hippocampus in one slice of each mouse, the same quantification for cortex site). **i** Immunofluorescence images of Aβ plaques, Iba-1, and GFAP in hippocampus and cortex regions of ex vivo brain slices, red (Aβ plaque, Iba-1, and GFAP) and cell nucleus (blue), all scale bars: 100 μm. **j, k** The corresponding fluorescence quantification of Iba-1 and GFAP from the hippocampus and cortex of the brain slices (3 biologically independent mice and respectively sampling two tested sites in the hippocampus in one slice of each mouse, the same quantification for cortex site). **l, m** The representative images and corresponding scoring comparisons of various groups in the nesting test (*n* = 8 biologically independent mice). **n** Swimming tracks of various groups. **o**–**q** were respectively the escape latency (**o**), the staying time in the target quadrant (**p**), and the crossing number through the platform (**q**) in the Morris water maze (MWM) test (*n* = 8 biologically independent mice). **r** Recognition index in the NOR test (*n* = 8 biologically independent mice). All dosages were 10 mg/kg. All data were presented as mean ± SEM. NS, no significance, *$P < 0.05$, **$P < 0.01$, ***$P < 0.001$, and ****$P < 0.0001$. All statistical analyses were compared by analyses of one-way ANOVA except the two-tailed Student's *t*-test in **e**. Source data are provided as a Source Data file.

from various groups; all morphologies indicated no obvious damage (Supplementary Fig. 35). Moreover, the levels of blood parameters of Ang-NCs treatment revealed no significant difference with PBS treatment, demonstrating good biocompatibility for in vivo AD theranostic (Supplementary Fig. 36).

## Discussion

Brain health is the most challenging frontier area in modern medicine. AD is one of the multifactorial neurodegenerative disorders. The toxic Aβ aggregation and inflammation are two reliable and accompany parallel representation. Unfortunately, there is no effective drug to satisfy the in vivo dual-target, brain-target, and visual theranostic requests despite tremendous investments. In this work, we have validated an original work, fabricating a brain-targeting, fibril-degrading, and ROS-regulating NIR-II nanotheranostic system via ingenious molecule design and co-assembly strategy, that successfully resolves the long-standing challenge in the AD field. To our delight, the high-sensitivity NIR-II imaging verified the Ang-2 modification assisted the NCs to efficiently cross the BBB. Triggered by the high ROS in the AD lesion, the NCs degraded and released **3** and **6**. **3** selectively inhibited the fibril generation, degraded the toxic Aβ fibril, and relieved the ROS as well as inflammation. **6** remodeled the cerebral redox balance and enhanced the above therapy effect of **3**, together reversing the neurotoxicity and achieving effective behavioral and cognitive improvements in the AD mice model. Such a powerful cascade of relay enhanced the therapy output, yielding improved behavior and memory in AD mice.

Benefited from the previous reports and our experiences, we summarized an instructive guideline of molecule design and nanostructure modulation of NIR-II AD nanotheranostic: (i) high-specificity to the aimed target is the primary consideration of all efficient and safe drugs, some special groups need to be grafted onto the probe, e.g., dimethylamino, diethylamino, or hydroxy groups, that were ideal groups specially bound with Aβ fibrils[42,43]. (ii) multi-competitive interactions were generally introduced for efficient degradation of Aβ fibrils, e.g., Van der Waals forces, π-π stacking, and hydrogen bonding. (iii) the coordination compound with reducing metal may be a high-efficiency antioxidation strategy for anti-inflammation. (iv) the high brightness, anti-quenching, and long-wavelength NIR-II emission are the future direction of drugs with the popularity of clinic NIR imaging, which need simultaneously balanced multi factors, e.g., large planar conjugation, large twisted structure, and multi-rotors. (v) the megamerger of high biocompatibility and superior BBB penetration. (vi) the smart nanodrugs responsive to the microenvironment of neurodegenerative diseases are future trends to acquire long in vivo half-life, maximize the drug availability in lesion regions, and minimize the toxicity to normal cells. These important guidelines are generalizable to the fabrication of NIR-II drugs against other neurodegenerative diseases. Our goal is to provide insight into the development of advanced NIR-II anti-neurodegenerative nanodrugs to guard brain health.

## Methods

### Ethical regulations

The animal experiments and protocols follow the guidelines of the Medical and Scientific Research Ethics Committee of Henan University School of Medicine (P. R. China, HUSOM 2018-354).

### Materials

Aβ peptide powder was purchased from Sigma-Aldrich Chemical Reagent Co., Ltd. Angiopep-2 peptide (TFFYGGSRGKRNNFK TEEYC) with thiol group was synthesized by China peptide Co., Ltd. (Suzhou, China). The amphiphilic polymer with various molecular weight and thioketal (TK) bond linkage-bridged DSPE-TK-PEG$_{2000}$, DSPE-TK-PEG$_{3400}$, DSPE-TK-PEG$_{3400}$-Mal, and iron oxide (Fe$_3$O$_4$) nanoparticles with 10 nm of diameter were obtained from Xi'an ruixi Biological Technology Co., Ltd. (China). The Fe$_3$O$_4$ nanoparticles (1 mg) need to be modified by DSPE-PEG (10 mg) under sonication and evaporation before use. Cell Counting Kit-8 (CCK-8) and 2',7'-dichlorodihydrofluorescein diacetate (DCFH-DA) were supplied by Beyotime Biotechnology (China).

### Co-assembly of Ang-AIE NCs

The NCs were prepared by co-assembly. Briefly, 1 mg **3** and 1 mg **6** were dissolved in N, N-dimethylformamide (100 μL). Then the mixture was dropped into 10 mL of deionized water containing 2 μmol of DSPE-TK-PEG$_{2000}$. Then the solution was sent to the probe sonication (80 W, on/off cycle was 5 s/5 s). Lastly, the resulting solution was centrifuged, washed, and dispersed in deionized water to acquire the NCs. The Ang-NCs were prepared using the same procedure with the mixture of 1.6 μmol of DSPE-TK-PEG$_{2000}$ with 0.4 μmol of DSPE-TK-PEG$_{3400}$-Ang instead of the primary DSPE-TK-PEG$_{2000}$.

### Preparation of Aβ monomer and fibril

The Aβ monomer was prepared through the dissolve and centrifugation procedures. Briefly, 0.1 mg of Aβ peptide powder was dissolved in 11 μL of dimethylsulfoxide (DMSO) solvent with water bath sonication for 10 min. Then 432 μL of PBS buffer (10 mM, pH = 7.4) was added into the above mixture and further sonicated for 10 min. The supernatant was collected as Aβ monomer after centrifugation at 14,000 × *g* for 15 min. The monomer solution was further incubated for 12 h and 48 h at 37 °C to obtain the oligomer and fibril samples, respectively.

### Degradation of Aβ fibril and restructuring of fragment

To verify the disassembly ability of the NCs against Aβ fibril, 1 mL of PBS solution containing fibril (25 μM) was co-incubated with NCs (200 μg/mL) for 48 h at 37 °C. Then the solution was separated and the Aβ products were collected after adding 10 μL of dimethylsulfoxide

(DMSO). The precipitate (fragment) was divided into two samples, one was supplemented with fresh Aβ monomer (final concentration: 25 μM) for the restructuring growth. Another was added the same amount of Aβ monomer containing NCs (final concentration: 200 μg/mL). Two samples were allowed to be co-incubated and grown for various times. Lastly, all samples were sent for TEM observation after staining with uranium acetate solution (1%).

## Detection of intracellular ROS

To test the in vitro antioxidation effect of NCs, DCFH-DA was used for the detection of intracellular ROS in PC12 cells with various treatments. In detail, the PC12 cells ($1 \times 10^5$ cells per well) were seeded in the microscope plate and grown overnight. Then these cells were incubated with various treatments including PBS, Aβ fibril (10 μM), and Aβ fibril (10 μM) with Ang-NCs (100 μg/mL) for 48 h at 37 °C. The cells were successively stained by the Hoechst 33342 and DCFH-DA (10 μM) according to the specification before confocal microscope observation.

## Computational methodologies

Assisted by the Gaussian 16 program package[53], the density functional theory (DFT) calculations were performed. The geometry optimizations of minima states involved were carried out at the B3LYP-D3 level of theory[54] with the 6-31 G(d) basis set. Solvation energies were computed at the B3LYP-D3 level of theory with the 6-311 + G (d,p) basis set using the gas-phase optimized structures and the SMD model[61] in water. The $S_1$ state was carried out with Time-Dependent Density Functional Theory (TD-DFT)[62]. The structure of Aβ and complex structures were solvated in a water box using AmberTools 20[52]. In predicting how **3** interferes with the Aβ protein, we started with the structure of Aβ and performed molecular dynamics (MD) simulations on the protein in water for 100 ns, which is long enough for equilibration. Then, we took the protein structure at the end of 100 ns and docked with the DFT-optimized structure of **3** using AutoDock Vina[55]. The MD simulations of complex structures were conducted using Amber 20 for 100 ns in water. **3** models constructed in water were used for MD simulations with RESP[63] charges. General Amber force field (GAFF)[56] was used for **3**. The FF14SB[57] force field was used for the proteins. The TIP3P force field was used for water molecule[58]. In water, the system was minimized for 20000 steps, gradually heated to 298 K and then equilibrated for 10 ps under constant volume and temperature. The production run was conducted for 100 ns under constant pressure and temperature. Fifty snapshots were taken in the whole 100 ns with 2 ns interval to form an ensemble of docking structures. The graphs of structures were generated using CYLview[64] and PyMOL (http://www.pymol.org/). GaussView 6.0.16 was used to construct the initial structures used in our computations. Schematic diagrams of the interactions between protein and **3** were shown by LigPlot[65]. The HOMO and LUMO were shown by Multiwfn[66] and VMD[67].

## Cell lines

The PC12, BV2, SH-SY5Y, and Neuro-2a cell lines were bought from Procell Life Science & Technology Co., Ltd. The bEnd.3 cell line was provided by the American Type Culture Collection (ATCC). The HT-22 cell line was obtained from Sunncell Biotech Inc. The PC12 cell line and Neuro-2a cell line were respectively cultured in the RPMI 1640 medium and MEM medium. The BV2, SH-SY5Y, HT-22, and bEnd.3 cell lines were grown in the DMEM medium.

## Reverse transcription-polymerase chain reaction

HT-22 cells ($2 \times 10^5$ cells per well) were cultured in the 6-well plates for 12 h. Then, the cells were respectively treated with PBS, Aβ fibril (10 μM), and Aβ fibril (10 μM) with Ang-NCs (100 μg/mL) for one day. Subsequently, the total RNA of all cells was extracted with FastPure®

Cell/Tissue Total RNA Isolation Kit V2 (Vazyme Biotech Co., Ltd). The cDNA was synthesized using the above RNA template (1 μg) and PrimeScript RT Master Mix (Takara). The RT-PCR analysis was performed by LightCycler 480II real-time detection system. The sequence of PCR primer (Tsingke Biotechnology Co., Ltd.) is shown as follows:

IL-6-F, AGCCAGAGTCCTTCAGAGAG;
IL-6-R, CTTAGCCACTCCTTCTGTGAC;
IL-1β-F, TGTGTAATGAAAGACGGCA;
IL-1β-R, TCCACTTTGCTCTTGACGGCAC;
TNF-α-F, CAAAATTCGAGTGACAAGCCT;
TNF-α-R, CTGGGAGTAGACAAGGTACAAC;
GAPDH-F, TTGATGGCAACAATCTCCA;
GAPDH-R, CGTCCCGTAGACAAAATGGT.

## Western blot assay

PC12 cells and HT-22 cells ($2 \times 10^5$ cells per well) were seeded in the 6-well plate, cultured overnight, and respectively incubated with PBS, Aβ fibril (10 μM), and Aβ fibril (10 μM) with Ang-NCs (100 μg/mL) for 24 h at 37 °C. The cells were treated with RIPA buffer after washing two times. Then the protein concentration was determined by a BCA kit. Then 15 μg of proteins were separated by 12.5% SDS-PAGE and transferred onto the PVDF membranes. After blocking with 5% defatted milk for 1 h at room temperature, the membranes were respectively incubated with Anti-GPX4 antibody (Abcam, ab125066, 1: 1000 dilution), Anti-iL-1β antibody (Proteintech, 26048-1-AP, 1: 1000 dilution), Anti-NLRP3 antibody (Hangzhou Huaan Biotechnology, ER1706-72, 1: 1000 dilution), and β-actin antibody (ABclonal AC026, 1: 50000 dilution) overnight at 4 °C. Lastly, the bands of these membranes were visualized with an ECL reagent and recorded by Super Signal ECL (Amersham Imager 680RGB) after 2 h of incubation with goat anti-rabbit IgG secondary antibody.

## In vitro cytotoxicity assessment

The PC12 cells ($5 \times 10^3$ cells per well) were cultured in 96-well plates. After 24 h, the cells were incubated with PBS, Aβ fibril (10 μM), and Aβ fibril (10 μM) with various concentrations of Ang-NCs at 37 °C for 48 h. Lastly, the CCK-8 detected agent was added to the wells and further incubated for 40 min. The optical density at 450 nm of the plates was recorded by a microplate reader (Devivces/13x).

## Animals

All AD mice in this work were seven-month-old female *APP/PS1* transgenic AD mice that were provided by Beijing HFK Bioscience Co. Ltd. The seven-month-old wild-type C57 mice and healthy 6- to 8-week-old Balb/c white mice were female and purchased from SiPeiFu (SPF) Biotechnology Co., Ltd. Mice are fed in Specific Pathogen Free (SPF) level laboratory animal center following standard experimental animal guidelines and ethics. The mice were caged in individually ventilated cages following a common light/dark cycle (12/12 h) at $22 \pm 2$ °C. The humidity is set as $55 \pm 10$%. The bedding in the cage was cleaned once weekly.

## Hemolysis assay

To evaluate the biocompatibility of Ang-NCs, we used a microplate reader to test the hemolysis ratio of erythrocytes. Briefly, the whole blood (300 μL) was harvested from Balb/c white mice and was placed in the anticoagulation tubes coated with ethylenediamine tetraacetic acid (EDTA). Then the samples underwent centrifugation at $845 \times g$ for 10 min at 4 °C and were washed five times with PBS. The resulting erythrocytes were dispersed in 5 mL of PBS. Subsequently, equivalent parallel dispersions (0.5 mL) were respectively incubated with 500 μL of deionized water, PBS, and various concentrations of Ang-NCs solution in PBS buffer at 37 °C for 2 h. Lastly, the supernatants of each sample were collected after centrifugation for the test of OD values at 570 nm.

## In vivo pharmacokinetics test

To assess the in vivo pharmacokinetics, the healthy Balb/c white mice (6–8-week-old, $n = 3$ biologically independent mice) were respectively i.v. injected with 200 μL of NCs and Ang-NCs (injection dosage is 10 mg/kg). We collected the blood at predetermined time points and subsequently centrifuged at $845 \times g$ for 15 min at 4 °C. Then 20 μL of supernatant was mixed with 80 μL of PBS in 96-well plates to measure the OD values at 365 nm. The OriginPro 2021 software was used to analyze the elimination half-life of two groups.

## In vivo NIR-II imaging

The seven-month-old female APP/PS1 mice were selected as AD mouse model. First, the Ang-NCs were i.v. injected into the AD mice (injection dosage is 10 mg/kg). The fluorescence imaging of cerebral vessels in mice without removing the scalp or skull was monitored at different channels with an NIRvana640 InGaAs camera under 808 nm irradiation.

## Behavior assessments on AD mice

All AD mice were randomly divided into different groups ($n = 8$ biologically independent mice) with one mouse per cage to avoid fighting. The AD mice were respectively i.v. injected with PBS and Ang-NCs (dosage: 10 mg/kg) once every three days. The total injection was six times. The WT mice also were *i.v.* injected with PBS six times.

Nest construction (NC): all mice were sent to the cage (one per cage). A pad of test napkin paper with three pieces was placed in the cage. The next day, the situation of paper in the cage was photographed and scored following scoring criteria[28].

Morris water maze (MWM): a pool was used as the test equipment and divided into four quadrants affixed with various symbols. The pool was filled with water containing titanium dioxide powder and kept at $22 \pm 1$ °C and exclude noise and strong light. The mice were trained daily for five consecutive days to seek the hidden platform. The time spent moving from the start site to the target platform (escape latency) was recorded in the first five days. After removing the platform, the moving route, crossing number, and staying time in the target quadrant were recorded on the sixth day.

Novel object recognition (NOR): the open box was used as the test apparatus. The tracking route was recorded by software to obtain manual behavior scoring. Briefly, the mice were first habituated to the empty tested box for 5 min. The next day, mice were trained to explore the pre-placed two identical objects for 10 min. On the third day, the same experiments were repeated except using a novel object to replace one of two original objects. The recognition index (RI) was calculated to assess the differences in exploration time among various groups.

## Assessment of antioxidant, anti-inflammation effect, and quantification of $A\beta_{1-42}$ level

To detect the antioxidant therapy effect of Ang-NCs, the brains of the APP/PS1 mice were harvested after i.v. injection of Ang-NCs one time. The DCFH-DA probe was dropped onto the surface of isolated fresh brains and incubated for 30 min. Then the fluorescence of ex vivo brains was observed under the equipment (IVIS Lumia III) post-dropping. To investigate the levels of inflammation factors and quantify $A\beta_{1-42}$ in the brain of mice after i.v. injection of Ang-NCs six times, the brain of mice was collected and homogenized for 4 min (70 Hz) in the 0.2 mL tube with RIPA buffer. Then the solution was centrifuged at $100,000 \times g$ for 30 min. Lastly, we tested the inflammatory mediators of the supernatants with the corresponding kits. The precipitate was treated with formic acid under sonication. The $A\beta_{1-42}$ in the solution was centrifugated and tested with an ELISA kit. The concentration of proteins was quantified by a BCA kit.

## Preparation and immunofluorescence staining of brain slices

The mice were anesthetized and perfused transcardially with saline after treatment. Then the brains of mice were harvested and fixed via immersion in paraformaldehyde for three days. After the dehydration treatment by gradient concentrations of alcohol, the brains were embedded into paraffin. The brain slices with 4 μm of thickness are prepared with a pathology slicer (Leica RM2016) and mounted onto glass slides. The slices were deparaffinized after the successive treatment of xylene, alcohol, and washing with deionized water. Then these slices were immersed in the EDTA antigen retrieval buffer (pH = 8.0) and sent to the microwave for 8 min. After being washed with PBS buffer (pH = 7.4), it was further incubated with bovine serum albumin (BSA, 5%) for 1 h and further incubated with the diluted primary antibody in PBS overnight at 4 °C. Lastly, the slices were washed and incubated with the secondary antibody for 1 h. Prior to the imaging observation using CaseViewer (Pannoramic MIDI), the slices were washed with PBS and then stained with DAPI (Servicebio G1012). The information of primary and secondary antibodies includes Rabbit pAb anti-Aβ (Servicebio, GB111197, 1: 400 dilution), GFAP (Servicebio, GB11096, 1: 500 dilution), Iba-1 (Servicebio, GB113502, 1: 500 dilution), 8-OHdG antibody (Santa Cruz, sc-393871, 1: 50 dilution), Goat anti-rabbit IgG (Servicebio, GB21303, 1: 300 dilution), and Goat anti-mouse IgG Alexa Flour 488 (Abcom, ab150117, 1: 500 dilution).

## Cerebral microbleeds observation

The microbleeds risk was investigated by the Prussian Blue histochemistry of brain slices. The slices were prepared according to the above procedure. Then the slices were treated with 2% potassium ferrocyanide in HCl (2%) in Coplin jars for 1 h at room temperature, followed by a counterstain in the Nuclear fast Red solution (0.1%) for 5 min. Subsequently, the blue haemosiderin-deposition profiles in the stained brain slices were imaged by Grundium OCUS40X.

## Aβ plaque detection of ex vivo brain slices staining by ThS

To investigate the level of in vivo Aβ plaque after the treatment, the ex vivo brain slices were respectively stained by Rabbit pAb anti-Aβ probe and ThS dye. First, the slice was immersed in the EDTA antigen retrieval buffer (pH = 8.0) and sent to the microwave for 8 min. After washing for three times with PBS buffer (pH = 7.4), it was treated with 5% BSA for 1 h and further incubated with the diluted primary antibody in PBS overnight at 4 °C. The slices were washed and underwent another incubation of secondary antibody for 1 h. Secondly, for the ThS staining, the new slice was stained with ThS dye for confocal observation. Briefly, the slices were stained with ThS solution (100 μM) in 50% ethanol for 20 min and washed with 80% ethanol for 10 s and water for 10 s. After sealing with an anti-fluorescence quenching agent, these slices were sent to the CLSM observation (Zeiss 880).

## Safety assessment

The in vivo toxicity of our NCs was investigated through histological observation and blood safety examination. Five main organs were harvested and prepared into H&E staining slices for microscopic observation after finishing behavior assessments. The whole blood was acquired from the healthy Balb/c mice (6–8-week-old, $n = 3$ biologically independent mice) on the eighteenth-day post i.v. injection of Ang-NCs (dosage: 10 mg/kg) and PBS six times. Then, these blood samples were performed for the routine blood test. The serums were sent for the biochemical analysis.

## Statistical analysis

All results are shown as mean ± standard deviation (SD) and means ± standard error of the mean (SEM). Statistical analysis among multiple datasets was compared by analysis of one-way analysis of variance (ANOVA), while that in two groups of dataset experiments was compared using a two-tailed Student's *t*-test. The reproducibility for all

experiments was performed according to at least three technical replicates.

## Reporting summary

Further information on research design is available in the Nature Portfolio Reporting Summary linked to this article.

## Data availability

All data generated or analyzed during this study are included in this published article its Supplementary Information file, and the Source Data file. The full image dataset is available from the corresponding author upon request. The accessible link for 5KK3 is https://www.rcsb.org/structure/5KK3. Source data are provided in this paper.

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

## Acknowledgements

This work was supported by the National Natural Science Foundation of China (NSFC 22275050 from J.W., 32001071 from J.W., and 22205096 from J.Z.), Joint Training Funds for Science & Technology R&D of Henan Province (222301420059, J.W.), Program for Science & Technology Innovation Talents in Universities of Henan Province (22HASTIT004, J.W.), the Outstanding Youths Development Scheme of Nanfang Hospital, Southern Medical University (2021J002, J.Z.), NHMRC Investigator Grant, and GuangDong Basic and Applied Basic Research Foundation (2021A1515110373, J.Z.), Natural Science Foundation of Henan Province (222300420113, X.W.). The Transwell model in Fig. 5c and partial figure elements were drawn in BioRender.com. The authors acknowledge the Beijing Super Cloud Computing Center (BSCC) for its high performance and AI computing resources, which contributed to the research results reported in this paper. URL: http://www.blsc.cn/.

## Author contributions

B.Z.T. and B.S. conceived the concept for this work. J.W. and P.S. contributed to the preparation and measurements of materials. X.C. and Y.Z. assisted with the characterizations. M.L., M.H., Y.L., L.H. and M.L. (Mengya Lu) assisted with the cell and animal assays. X.P. assists with the behavior test. X.W., Y.L. (Yang Liu), and C.S. helped to analyze the data. Y.Z. (Yuan Zhou), H.Y., J.C. and X.W. (Xin Wang) carried out the binding calculations of proteins. J.Z. synthesizes the compounds. B.S., J.W. and X.W. (Xin Wang) wrote the manuscript.

## Competing interests

The authors declare no competing interests.
