## [Peer Review File · Nature Communications]

Reviewers' Comments:

Reviewer #1:

Remarks to the Author:

This manuscript by Wang et al, has described in detail chemistry and synthesis of Aggregate Induced Emission probes based nanocomposites that inhibit Abeta fibrillation, break the pre-formed fibrils and inhibit the associated ROS and inflammation. The two probes, i.e., PTB and TPTB-Ce perform the breakdown of Abeta fibrils and ROS mitigation, respectively. The idea and design of the nanoparticles are smart. The manuscript is interesting and thorough characterisations are performed. Here are some suggestions that need to be addressed before the manuscript is suitable for publications.

1. The ROS inhibition is rationally associated with Ce chelates in the TPTB complex, however, the structure of the molecule looks polyphenolic and polyphenolic compounds generally have anti-oxidation behavior. Can it be distinguished?
2. The NCs claimed to inhibit the reaggregation of the Abeta fibrils. Fig. 3I show restructuring of the broken fibril back into longer aggregates by adding PBS. How is this feasible? By adding PBS, the solution of Abeta fragments and NCs will be diluted and there will be less chance for them to remake fibrils. Also, any reported case of remaking of the broken fibrils?
3. The toxicity and cellular interaction of the NCs induced degraded products of Abeta should be studied.
4. In response to H₂O₂, the particle size is increased. If particles are releasing/disintegrating PTB and TPTB-Ce, then particle size should reduce. It should be confirmed with TEM that what happens to the particles morphology when exposed to H₂O₂.
5. The TEM of monomers present smaller aggregates. Were monomers already in aggregated form?
6. Supplementary 17. The ThT, ThS and PiB do not inhibit the fibrosis. They just provide kinetics data on the fibrosis. Also, the authors should include the ThT experiment on the inhibition kinetics of NCs against Abeta monomers.
7. Fig. 5b. The NCs take 6 hrs to internalize while in vivo elimination half-life is 4.8 hrs. Can authors explain how in vitro and in vivo difference could be co-related. The particles will be eliminated before internalization.
8. Page 9, authors mention the best NIR images optical density is from kidney and liver but fig. 6d present liver and spleen.

Minor:

1. Include the details on the mice models used (APP/PS1) in the results and discussion sections. The AD mice models does not accurately present the models used.
2. Caption of fig.2. There must be a word after cardiovascular? Such as system, or vessels.
3. Page 1, "binding with Abeta plaques of the drugs" is confusing. Does it mean binding of drugs with Abeta plaques?
4. The title looks like a brief description of the results. It could be changed to a better version.

Reviewer #2:

Remarks to the Author:

Wang et al described a novel NIR II- aggregation-induced emission (AIE) nanotheranostic theranostic probe targeting at amyloid-beta and ROS in APP/PS1 Alzheimer's disease mouse models. The treatment improved the performance of APP/PS1 mice in several behavioural tests. The study is well-designed and includes extensive data. The manuscript is well-written and structured. However, the number of animal is very low for a treatment study (n=3 per group); further methodological details and a larger group size of animal need to be provided to draw conclusions. I would recommend major revision for this manuscript.

1. Please provide information on the sex of the animals, for both APPPS1 mice and Balb/c mice. Sex is an important factor in Alzheimer's disease, affecting the pathology level and neuroinflammation.
2. Please double check the age of animal. Figure 6 states 6 month APPPS1 while on page 11 states 7 month. Please provide more details on the study design (if it is randomized etc). the number of animal n=3 per group is very low for a treatment study;
3. Please provide logP and further information on the probe used for treatment.

4. Please include a dose escalation design for the treatment study. It is not clear why 10 mg/kg is the optimal dosage for treatment.
5. What is the mechanism of amyloid plaque reduction after the treatment using the probe. Is it involves macrophage-mediated clearance. Fig 6p Iba1 fluorescence staining results, only showed a reduction in the levels of Iba1+ microglia in the hippocampus. Please include further staining to clarify this. Please include staining on cortical region as well in the results.
6. The author stated that it is microscopic observation of A β fibril on ex vivo brain slices (Fig. 6h). Which antibody or dye was used. Is it fibril conformation specific? What about small forms of A β aggregates such as A β oligomers which are known to be more toxic than A β fibril. Please include further staining data and/or biochemical data to clarify this.
7. Method details on brain extraction, brain slices, immunofluorescence staining and antibody information (for A β , GFAP, and Iba1 staining) is missing.
8. Is there any consequences such as microbleeding that commonly observed in amyloid plaque clearance studies. Please provide histology or MRI evidence on this aspect.
9. Toxicity test, It is stated that the healthy Balb/c mice were injected with 200 μ L of Ang-NCs (dosage: 10 mg/kg) and PBS. The whole blood was collected from the orbital of the mice on day 10 post the injection. However given that the treatment is 6 time injection over 2 weeks in APP/PS1 mice, a similar design is needed in the tox study to reflect the potential influence. Please include further supporting data on this.

Reviewer #3:

Remarks to the Author:

The authors described a novel nanotheranostic system that targets the brain, with fibril-degrading and ROS-scavenging effects. They showed that the system was BBB permeable and was neuroprotective in vitro and in vivo. In vitro, the system protected cultured PC12 and HT-22 cells against A β fibrils. In AD model mice, the system not only reduced A β load, neuroinflammation and oxidative stress, but also improved functional outcome.

1. The authors claimed this was nanotheranostic system. While they showed its therapeutic application, they did not demonstrate any usefulness in imaging or diagnostic applications. As such, the rationale for designing a NIR-II probe was not justified.
2. For the therapeutic application, they provided a lot of data from in vitro and in vivo. But experimental details are generally lacking, making it difficult to interpret the quality of data. For example, they used APP/PS1 transgenic AD mice in the in vivo experiments. The mice were 7-month-old when they were obtained, but it is not clear when the treatment started, and when the behavioral tests were performed. In addition, n=6 mice/group is a little too small for behavioral tests.
3. Another major concern is lack of control. Most of the in vitro and in vivo experiments were conducted without proper control.
4. They showed A β fibril staining after treatment (Fig 6h), however, no quantitative data were provided. Similarly, no quantitative data for the immunofluorescence of Iba1 and GFAP. In addition, to show the A β fibril-reducing effects, the authors need to provide additional evidence, with another measurement of A β , for example, western or ELISA.
5. Similarly, the evidence of in vivo anti-oxidative stress is rather weak (Fig 6f). It was from ex vivo brain imaging (again no mention of how this was done in the "Methods" sections). There are better ways to assess oxidative damage.
6. In Fig 6c,d, the scale bars seem incorrect (the dark end represents higher signal?). Also, the images from Fig. 6d don't support the observation of "a major accumulation in the liver and kidney". The strength of signals in the liver and kidney are completely different.
7. Minor. The author seemed to use "fibrosis" to describe the formation of A β fibrils. This is not common in the AD field, since the term "fibrosis" usually refers to the accumulation of fibrous connective tissue.

List of changes: Point-to-Point Responses to the Comments of Reviewers

Reviewer: 1

Recommendation: Reconsider after major revisions noted.

Comments: *This manuscript by Wang et al, has described in detail chemistry and synthesis of Aggregate Induced Emission probes based nanocomposites that inhibit Abeta fibrillation, break the pre-formed fibrils and inhibit the associated ROS and inflammation. The two probes, i.e., PTB and TPTB-Ce perform the breakdown of Abeta fibrils and ROS mitigation, respectively. The idea and design of the nanoparticles are smart. The manuscript is interesting and thorough characterisations are performed. Here are some suggestions that need to be addressed before the manuscript is suitable for publications.*

- 1) The ROS inhibition is rationally associated with Ce chelates in the TPTB complex, however, the structure of the molecule looks polyphenolic and polyphenolic compounds generally have anti-oxidation behavior. Can it be distinguished?

Response: Thanks for your kind suggestion. The polyphenolic compounds generally have one or more aromatic rings and phenolic hydroxy groups with acidic character for anti-oxidation activity. Differently, our TPTB-Ce complex is a polymer (**Fig. 2a**), and PTB has no above characteristic groups. Therefore, we attribute the main contribution of anti-oxidation behavior to the Ce active center, not PTB or the organic backbone from TPTB-Ce. Because the control molecule for TPTB-Ce can't exist without Ce center, we employed more assays to identify the antioxidant function of TPTB-Ce. First, the laser confocal microscopy result indicated strong green fluorescence of A β fibrils appeared in the TPTB-Ce with A β fibrils group and A β fibrils group. However, it greatly faded in the PTB with A β fibrils group (**Supplementary Fig. 25a**), suggesting the effective disassembly capability against A β fibril was generated from PTB, not TPTB-Ce. Subsequently, we further detected the antioxidant ability of two molecules using a reactive oxygen species (ROS) probe (DCFH-DA). The quantification of oxidative product (DCF) indicated the strong antioxidant ability of TPTB-Ce, which greatly exceeds PTB (**Supplementary Fig. 25b**). Lastly, the CCK-8 assay result indicated both two molecules have the superior neuroprotective function to reverse neurotoxicity induced by A β fibrils and excessive oxidation (**Fig. 5I**). The additional results combined with massive original results together demonstrated the definite therapy contribution, i.e., PTB is responsible for the degradation of A β fibrils and TPTB-Ce for anti-oxidation, together reduced the neuron damage. We added the necessary explanation for the polymer properties of the TPTB-Ce complex in the text that was marked red in Line 9, Page 3 “**The ends of TPTB-Ce are terpyridyl coordinated Ce(III) units, which endows TPTB-Ce with effective antioxidation capability⁴⁴⁻⁴⁶. Actually, TPTB-Ce is a polymer according to the dual coordination mode of Ce(III) centers, but not a small molecule**”.

Fig. 2 a Chemical formula of PTB and TPTB-Ce (the filling color was deleted for better recognition).

Supplementary Fig. 25 a Confocal images of PC12 cells incubated with PBS, A β fibril, A β fibril with TPTB-Ce, and A β fibril with PTB for 48 h. All cells were stained with ThT (green fluorescence). **b** The ROS level of three samples tested by the microplate reader and stained with the DCFH-DA probe. The concentrations of two AIEgens and A β fibril were respectively 100 μ g/mL and 10 μ M.

Fig. 5 I Cell viability of PC12 cells treated with A β fibril, A β fibril with TPTB-Ce (100 μ g/mL), A β fibril with PTB (100 μ g/mL) for 48 h. The concentration of A β fibril was 10 μ M.

2) The NCs claimed to inhibit the reaggregation of the Abeta fibrils. Fig. 3I show restructuring of the broken fibril back into longer aggregates by adding PBS. How is this feasible? By adding PBS, the solution of Abeta fragments and NCs will be diluted and there will be less chance for them to remake fibrils. Also, any reported case of remaking of the broken fibrils?

Response: Thanks for your kind suggestion. The restructuring was performed by replenishing the PBS containing A β monomer. We are sorry for the lack of marking of containing A β monomer following PBS in the picture. Therefore, we revised the illustration in **Fig. 3e** and respectively used "Fragment + monomer" and "Fragment + monomer + NCs" instead of the original "Fragment + PBS" and "Fragment + NCs" for more accurate understanding by the reader (**Fig. 3i**). The assay has been repeated and got a similar morphology change trend to the original image in the text: the morphologies of fragments gradually changed to the amorphous particle at 12 h, entanglement at 24 h, and fibril at 48 h, while it could be greatly inhibited by the NCs (**Fig. R1**). The restructuring phenomenon was only simply mentioned in a few reported cases, such as the fourth restructuring mode as shown in **Fig. R2** (*Nature Protocols* 2016, 11, 252), the primary and secondary nucleation and also fragmentation as seeds could induce the growth of adding A β monomers on to the ends of growing fibrils, which leads to the broken fragment back

into longer aggregated fibrils and mass increases. Another reported case (*Scientific Reports* 2018, 8, 16190, DOI:10.1038/s41598-018-33935-5) also claimed that the number of aggregates can increase by both primary and secondary nucleation and fragmentation. The growth phase (elongation) is dominated by the addition of A β monomers onto the ends of growing fibrils, which leads to rapid increases in fibril mass. The detailed experimental procedures were added in the Methods section and shown as follows:

Degradation of A β fibril and restructuring of fragment

To verify the disassembly ability of the NCs against A β fibril, 1 mL of PBS solution containing fibril (25 μ M) was co-incubated with NCs (200 μ g/mL) for 48 h at 37 $^{\circ}$ C. Then the solution was separated and the A β products were collected by centrifugation at 12,000 g for 20 min after adding 10 μ L of dimethyl sulfoxide (DMSO). The precipitate (fragment) was divided into two samples, one was supplemented with fresh A β monomer (final concentration: 25 μ M) for the restructuring growth. Another was added the same amount of A β monomer containing NCs (final concentration: 200 μ g/mL). Two samples were allowed to be co-incubated and grown for various times. Lastly, all samples were sent for TEM observation after staining with uranium acetate solution (1%).

Fig.3 e Illustration for the experiment design of oxidation promoted the formation of A β fibrils, and the fibrils inhibition, fibrils degradation, and reaggregation inhibition by NCs.

Fig. 3 i TEM images of re-aggregation of fragments and re-aggregation inhibition by NCs.

Fig. R1 The TEM images on a large field of fragments at 12 h, 24 h, and 48 h post adding A β monomer, and the fragment treated with Ang-NCs for 48 h with adding the same amount of A β monomer.

Model names	Scaling	Curvature in $t_{1/2}$ Plots	1° Nucleation	Elongation	Elongation	Fragmentation	2° Nucleation	2° Nucleation
			k_n, n_c	1 step k_+	2 step k_+, k_E	k_-	1 step k_2, n_2	2 step k_2, n_2, K_M
Nucleation elongation*	$-\frac{n_c}{2}$	None	✓	✓				
Secondary nucleation**	$-\frac{n_2 + 1}{2}$	None	✓	✓			✓	
Fragmentation	$-\frac{1}{2}$	None	✓	✓		✓		
Fragmentation and 2° nucleation	$-\frac{1}{2}$ to $-\frac{(n_2 + 1)}{2}$	-ve	✓	✓		✓	✓	
Multistep 2° nucleation	$-\frac{(n_2 + 1)}{2}$ to $-\frac{1}{2}$	+ve	✓	✓			✓	✓
Saturating elongation	$-\frac{n_c}{2}$ to $-\frac{(n_c - 1)}{2}$	+ve	✓	✓	✓			
Saturating elongation and 2° nucleation	$-\frac{(n_2 + 1)}{2}$ to $-\frac{n_2}{2}$	+ve	✓	✓	✓		✓	
Saturating elongation and fragmentation	$-\frac{1}{2}$ to 0	+ve	✓	✓	✓	✓		

*Oosawa and Asakura **Ferrone et al.

Fig. R2 Mechanisms and models. A list of all the different mechanisms of aggregation considered in this protocol is given, together with the models from which they are derived. Many of the simpler models are limits of the more complicated ones— e.g., nucleation elongation is a special case of all other models, when secondary processes become negligible.

- 3) The toxicity and cellular interaction of the NCs induced degraded products of Aβ should be studied.

Response: Thanks for your kind suggestion. The amyloid fibrils and metastable Aβ oligomers are thought to be two toxic species in Alzheimer’s disease (*Nat. Commun.* 2021, 12, 4634) via causing synaptic dysfunction including the overload of Ca ion, and activation of caspases cascade cases. However, the degraded Aβ products in our system are amorphous fragment rather than oligomer, which theoretically have no toxicity to cells. To verify the speculation, we studied the toxicity of degraded Aβ products (fragment) by NCs in PC12 and HT-22 cells. The cell viability of PC12 and HT-22 cells treated with oligomer were 38.6% and 34.2%, that in the fibril group were 54.5% and 48.7%, while 93.4% and 99.6% in the Aβ fragment group (**Fig.5m**), indicating negligible toxicity and certifying our speculation that the Aβ fragment is a safe species, not a toxic species.

Fig. 5 m Cell viability of PC12 and HT-22 cells incubated with Aβ oligomer (10 μM), Aβ fibril (10 μM), and degraded Aβ fragment (10 μM).

- 4) In response to H₂O₂, the particle size is increased. If particles are releasing/disintegrating PTB and TPTB-Ce, then particle size should reduce. It should be confirmed with TEM that what happens to the particles morphology when exposed to H₂O₂.

Response: Thanks for your kind suggestion. In previous reports, the decomposition of

responsive polymer NPs would generate two size change trends accompanied by cargo release, i.e., time-dependent size increase and decrease, both were reasonable. We speculate it was generated may dependent on the proportion of hydrophilic and hydrophobic groups left after broking. Such as in our previous work (*Nat. Commun.* 2022, 13, 6835), the hydrogel gradually decomposed and degraded to amorphous NPs with decreased size (**Fig. R3**) under irradiation may due to increased ratio of exposed hydrophilic group. However, the size increased also appeared in our previous works (*Adv. Mater.* 2019, 31, 1903277; *Adv. Sci.* 2023, 10, 2206333; *Biomaterials* 2021, 276, 121036; *Biomaterials* 2022, 289, 121760). The sizes were increased may be induced by the increasing proportion of exposed hydrophobic thiol group (ROS-responsive cases) or other groups (acid-responsive cases) after decompose (**Fig. R4**). We re-tested the DLS monitoring of NCs as a function of time on immersing in H₂O₂, which were sonicated for 2 s in water bath before testing (**Fig. 3a**). The result also indicated an obvious size increase of NCs as time increasing. Moreover, the TEM images of NCs immersed in H₂O₂ for 30 min indicated the NCs were swelled and broken into amorphous NPs (**Supplementary Fig. 16**) accompanied by appearing some black spots labeled by yellow arrows, demonstrating effective decompose.

Fig. R3 a,b The TEM images and sizes of the NGs@TMZ/ICG before (a) and after (b) NIR irradiation (808 nm, 0.5 W/cm², 5 min).

Fig. R4 a Size distribution graph and corresponding TEM image of TBTP-Au NPs before and after H₂O₂ treatment for 24 h (*Adv. Sci.* 2023, 10, 2206333). **b** Schematic illustration and size change of 3I-NM@siRNA following H₂O₂ treatment as determined by DLS (*Adv. Mater.* 2019, 31, 1903277). **c** The pH-induced disassembly of the NPs at pH 5.0 was determined by DLS at various time points. (*Biomaterials* 2022, 289, 121760). **d** Size, polydispersity index (PDI), and morphology change of Ang-NM@miRNA under high ROS environment (H₂O₂ treatment). (*Biomaterials* 2021, 276, 121036).

Fig. 3 a Size change comparison of NCs dispersed in PBS and hydrogen peroxide (H₂O₂).

Supplementary Fig. 16 a-c TEM images of the NCs before (a) and after H₂O₂ treatment for 30 min (b). c The magnified image corresponding to the white dashed frame in b. The image a was placed to the text as the inset in Fig. 2d.

5) The TEM of monomers present smaller aggregates. Were monomers already in aggregated form?

Response: Thanks for your kind suggestion. The TEM image of the A β monomer was reported as two main morphologies including no any species in the left image of Fig. R5 (*Jove-Journal of Visualized Experiments* 2018, e57592, doi: 10.3791/57592) or smaller aggregates in the right image of Fig. R5 (*Front. Pharmacol.* 2023,14,1103012). The difference was induced by different dilutions, i.e., the concentration of A β monomer in left image and right image were respectively 10 μ M and 50 μ M, that was 25 μ M in our system. Our result is closer to the right image with smaller aggregates by 50 μ M. We repeated the assay and the resulting morphology of the monomer still presented the same trend (Fig. 3f) as the previous image. We speculated the phenomenon may be caused by the unavoidable slight aggregation of high concentration of A β monomer with strong hydrophobic force during naturally drying process.

Fig. R5 TEM images of A β monomer with various concentrations (left: 10 μ M, right: 50 μ M). The left scale bar was 200 nm.

Fig. 3 f, TEM image of A β monomer.

- 6) Supplementary 17. The ThT, ThS and PiB do not inhibit the fibrosis. They just provide kinetics data on the fibrosis. Also, the authors should include the ThT experiment on the inhibition kinetics of NCs against Abeta monomers.

Response: Thanks for your kind suggestion. The fluoresce of ThT at 490 nm was used to monitor the inhibition kinetics of NCs against the A β monomer. The fluoresce intensity is directly related to the concentration of A β fibrils present. The corresponding result and explanation were marked red and added in the new manuscript as “Moreover, the ThT experiment on the inhibition kinetics of NCs against A β monomers was tested. The results indicated the 50% inhibiting concentration (IC₅₀) value of NCs is 9.67 μ g/mL, suggesting effective inhibition of NCs against the aggregation of A β monomer (Supplementary Fig. 21)” in Line 32, Page 5.

Supplementary Fig. 21 a Relative fluorescence intensity of ThT (5 μ M) in the mixture of A β monomer (10 μ M) with various concentrations of PTB/TPTB-Ce NCs. **b** Inhibition kinetic curve and IC₅₀ value for PTB/TPTB-Ce NCs against A β monomer.

- 7) Fig. 5b. The NCs take 6 hrs to internalize while in vivo elimination half-life is 4.8 hrs. Can authors explain how in vitro and in vivo difference could be co-related. The particles will be eliminated before internalization.

Response: Thanks for your kind suggestion. In this study, Ang-NCs were thought of as a drug. In general, the incubation time of the drug in cells was 24 ~ 48 h or longer time (72 h). Therefore, the internalized time in most of our in vitro assays including toxicity assays, confocal observation for DCF, MitoSOX, and JC-1, Western blot, ELISA, rtPCR, crystal violet staining, and BioTEM observation were selected for 48 h (not 6 h) due to the mutual interaction between NCs with fibril at cellular level after a gradual decompose. The internalized time is that the NCs spends on the interaction with cells, while in vivo elimination half-life is the time spent on the complicated interaction of NCs with vessels, accumulation/metabolism in main organs (heart, liver, spleen,

lung, and kidney) and BBB penetration. The two time values have no effect on each other and no direct relation. A similar difference between the two time values often appeared in the previous works of our or other groups (**Table R1**).

Table R1. The comparison between in vitro internalized time and in vivo half-life.

Num.	In vitro internalized time (hrs)	In vivo half-life (hrs)	References
1	48	8.1	Nat. Commun. 2022,13,6835
2	48	7.84	Adv. Mater. 2022, 34, 2203958
3	48	7.06	ACS Nano 2023, 17, 5, 5033
4	72	4.9	Nat. Commun. 2023,14,4557
5	48	2.68	Adv. Funct. Mater. 2020,30, 1910691
6	24	2.7	Adv. Mater. 2021, 34,2106082
7	24	1.86	ACS Nano 2022, 16, 11, 19053
8	24	1.75	Nat. Commun. 2021, 12, 3393
9	24	0.97	Nat. Commun. 2022,13,2834
10	72	0.95	Sci. Adv. 2022, 8, eabm8011
11	48	4.8	Our work

- 8) Page 9, authors mention the best NIR images optical density is from kidney and liver but fig. 6d present liver and spleen.

Response: Thanks for your kind suggestion. There is a difference between the NIR fluorescence intensity and the optical density (OD) quantification of homogenate, i.e., the high level of nanoparticles distributed in the liver and spleen in the former, and the liver, kidney, and spleen for the latter. We speculated it may be generated by two reasons: one reason was the difference of the test wavelengths and test mode, i.e., absorption at 720 nm and emission at 1350 nm. The other reason may be induced by the background value is not deducted, i.e., the OD values were the absolute OD without considering the high self-absorption of organs in the visible region. Therefore, we simultaneously re-tested the OD at 720 nm of homogenate from healthy C57 mice and the APP/PS1 mice injected with NCs and Ang-NCs treatments (n = 6), the result confirms the above hypothesis (**Fig. R6a**), revealing high self-absorption background at 720 nm tested by a microplate reader in healthy mice. The liver, spleen, and kidney of APP/PS1 mice have higher accumulations than the other organs. Differently, the NIR imaging eliminates the above interference of the visible background and could reflect the distribution level more accurately. As shown in **Fig. R6b**, the various organs of healthy mice treated with NCs and Ang-NCs revealed main accumulation in the liver and spleen, while the healthy mice injected with PBS have no signal, suggesting a high signal-to-noise ratio of NIR imaging. Therefore, to acquire a more accurate accumulation information of Ang-NCs, the fluorescence intensity of NIR imaging was used to quantify the distribution of two NCs in different organs and increased the number of APP/PS1 mice from 1 to 4. The results indicated both two nanocomposites were mainly distributed in the liver and spleen (**Fig. 6d, e**). The similar biodistribution result was common in the previous works shown in **Fig. R7b** (*Nat. Commun.* 2021, 12, 3393).

Fig. R6 a Optical density at 720 nm of various organs in APP/PS1 mice injected with NCs and Ang-NCs (dosage: 10 mg/kg), and healthy mice treated with PBS (n = 6). **b** NIR image at 1350 nm of various organs from the healthy mice treated with Ang-NCs (dosage: 10 mg/kg), NCs (dosage: 10 mg/kg), and PBS.

Fig. 6 d, e NIR images and corresponding fluorescence quantification for main organs of mice treated with NCs and Ang-NCs (n = 4), heart (H), liver (Li), spleen (S), lung (L), and kidney (K).

Fig. R7 b Distribution of Si element in main organs (hearts, livers, spleens, lungs, and kidneys) and tumors of mice at varied time points after FHPG administration.

Minor:

- 1) Include the details on the mice models used (APP/PS1) in the results and discussion sections. The AD mice models does not accurately present the models used.

Response: Thanks for your kind suggestion. All expression of AD mice has been revised to APP/PS1 mice for better understanding and we added the illustrations in “All AD mice in this work were seven-month-old female APP/PS1 transgenic AD mice that were purchased from Beijing HFK Bioscience Co. Ltd. (Beijing, China). The wild-type C57 mice and healthy Balb/c white mice were female and purchased from SiPeiFu (SPF) Biotechnology Co., Ltd. (Beijing, China)” in the Methods section and the legend “Fig. 6 | In vivo theranostic of Ang-NCs in

APP/PS1 (AD) mice”, which was marked by red.

- 2) Caption of fig.2. There must be a word after cardiovascular? Such as system, or vessels.

Response: Thanks for your kind suggestion. The cardiovascular has been revised to “**cerebral vessels**” in Caption of **Fig.2i**, which was marked by red.

- 3) Page 1, “binding with Abeta plaques of the drugs” is confusing. Does it mean binding of drugs with Abeta plaques?

Response: Thanks for your kind suggestion. The original expression has been revised to “**binding process of drugs with A β plaques**”, which was marked by red.

- 4) The title looks like a brief description of the results. It could be changed to a better version.

Response: Thanks for your kind suggestion. The original title “Brain-targeting, fibril-degrading and ROS-regulating NIR-II nanotheranostic system for Alzheimer’s disease” has been optimized to “**A one-two punch targeting reactive oxygen species and fibril for rescuing Alzheimer’s disease**”, which was marked by red.

Reviewer #2:

Wang et al described a novel NIR II- aggregation-induced emission (AIE) nanotheranostic theranostic probe targeting at amyloid-beta and ROS in APP/PS1 Alzheimer's disease mouse models. The treatment improved the performance of APP/PS1 mice in several behavioural tests. The study is well-designed and includes extensive data. The manuscript is well-written and structured. However, the number of animal is very low for a treatment study ($n=3$ per group); further methodological details and a larger group size of animal need to be provided to draw conclusions. I would recommend major revision for this manuscript.

1) Please provide information on the sex of the animals, for both APP/PS1 mice and Balb/c mice. Sex is an important factor in Alzheimer's disease, affecting the pathology level and neuroinflammation.

Response: Thanks for your kind suggestion. According to the reports (*Neurotoxicity Research*, 2016, 29, 256; *Journal of Neuroscience Research*, 2017, 95, 671; *Communications Biology*, 2021, 4, 711), The age, genetics, and sex are crucial to the development of Alzheimer's disease (AD). The female APP/PS1 mice are more suitable for AD research due to higher levels of A β burdens and microbleeds, and massive astrocytes and microglia than male mice. Therefore, the female APP/PS1, C57, and Balb/c mice were selected as the aimed mice and the missing detailed information was added in the Methods section and marked red "All AD mice in this work were seven-month-old female APP/PS1 transgenic AD mice that were purchased from Beijing HFK Bioscience Co. Ltd. (Beijing, China). The wild-type C57 mice and healthy Balb/c white mice were female and purchased from SiPeiFu (SPF) Biotechnology Co., Ltd. (Beijing, China)".

2) Please double check the age of animal. Figure 6 states 6 month APP/PS1 while on page 11 states 7 month. Please provide more details on the study design (if it is randomized etc). the number of animal $n=3$ per group is very low for a treatment study.

Response: Thanks for your kind suggestion. Both ages of all APP/PS1 and wild-type (WT) mice were 7 months. The inaccurate expression has been revised. Moreover, the expression "All AD mice were randomly divided into different groups ($n = 8$) with one mouse per cage" was added in the Methods section. All assays ($n = 3$) were redone with increased parallel samples, which includes all behavioral experiments ($n = 8$, **Fig. 6m-r**) including nest construction (NC) test ($n = 8$), Morris water maze ($n = 8$), and Novel object recognition ($n = 8$). The ELISA assay ($n = 5$, **Fig. 6g**). Moreover, we also increased the sample capacity of mice in biodistribution assay by NIR image from one to four mice ($n = 4$, **Fig. 6d, e**). The new results were listed as follows:

Fig. 6 g The levels of inflammatory factors (TNF- α and IL-1 β) and A β ₁₋₄₂ in the WT mice, APP/PS1 (AD) mice, and AD mice treated with Ang-NCs ($n = 5$).

Fig. 6 **m** The corresponding scoring comparisons of various groups in the nesting test (n = 8). **o-q** were respectively the escape latency (**o**), the time in the target quadrant (**p**), and the crossing number through the platform (**q**) in the Morris water maze (MWM) test (n = 8). **r** The recognition index in the NOR test (n = 8).

Fig. 6 **d, e** NIR images and corresponding fluorescence quantification for main organs of mice treated with NCs and Ang-NCs (n = 4), heart (H), liver (Li), spleen (S), lung (L), and kidney (K).

3) Please provide logP and further information on the probe used for treatment.

Response: Thanks for your kind suggestion. The calculated LogP (cLogP) was a common method to acquire the log P value (*Angew. Chem. Int. Ed.* 2023, 62, e202211550; *Neuro Rx.* 2005, 2, 541; *J. Med. Chem.* 2006, 49, 7559). The cLogP of PTB was calculated cLogP = 7.635 by Chem 3D 20.0 software, that was added to the text and marked by red. However, the TPTB-Ce has no logP due to it is a polymer not a small molecule. In general, the LogP value has important influence on the lipophilicity and BBB penetration of small drug molecules (*Mol. Imaging Biol.* 2023 doi: 10.1007/s11307-023-01843-4). As we known, the hydrophobic force and hydrogen bond were main interactions in A β aggregation (*Chem. Bio. Chem.* 2021, 22, 2306; *Nanoscale* 2018, 10, 8989). In previous system, to acquire high hydrophilicity and solubility, and suitable lipophilicity to increase the potential for BBB penetration, the clogP values of ideal central nervous system (CNS) drug molecules need to be selected a low range from 2 to 5, which

inevitably impaired the key hydrophobic force and hydrogen bond of molecules bound with A β fibril. In this work, we do not need to consider the restriction of cLogP to the BBB penetration because the Ang-2 assisted the nanoparticles traverse the BBB, therefore we appropriately increase the LogP value to 7.635 to increase the hydrophobic force of PTB and acquire higher binding capability. The calculated results in **Fig. 4** also verified the correctness of the above molecule design of PTB possessing Van der Waals forces, π - π stacking, and hydrogen bonding. Moreover, as illustrated in text, some special groups including diethylamine and hydroxy groups (*Chem. Sci.* 2016, 7, 4600; *J. Med. Chem.* 2015, 58, 6972), which were used for treatment also were grafted onto the PTB probe and were ideal moieties to specially bound with A β fibrils and competed with the self-aggregation between A β monomer for degradation of A β fibrils. Some of the above explanation has been added to the text for better understanding.

- 4) Please include a dose escalation design for the treatment study. It is not clear why 10 mg/kg is the optimal dosage for treatment.

Response: Thanks for your kind suggestion. In generally, to find the optimal dosage and following the "3R (Replacement; Reduction; and Refinement) principle" in animal experiments, i.e., using the minimum required number of experimental animals to verify whether the drugs have therapy effect, the researchers generally firstly investigate the concentration-dependent therapy efficiency on cellular level, then the resulted concentration was converted to the optimal in vivo dosage based on body weight of mice. In our system, the toxicity result of HT-22 cells and PC12 cells treated with various concentrations (25, 50, 100, 200, and 400 μ g/mL) of Ang-NCs indicated a gradually increased cell viability as a function of concentration. The 200 μ g/mL is an ideal concentration, which simultaneously maximized the neurotoxicity reverse of A β fibril and minimized the dosage-induced safety risk for in vivo assays (**Fig. 5k**). According to the weight (20 + 0.5 g) of nude mice, the in vivo dosage was set to 10 mg/kg, i.e., 200 μ g per mice. All in vivo results in **Fig. 6** indicated the effectiveness and high safety of the dosage, the NCs effectively degraded the toxic A β fibril and relieved the inflammation to remodel the cerebral redox balance, and reverse the neurotoxicity without inducing any toxic and side effect. This dosage is also commonly seen in the nanodrugs system of other research groups (**Table R2**). In next further works, we would pay more attention to the influence of concentration on the optimal therapy output as much as possible under observing "3R principle" principle besides confirming the therapy capability.

Fig. 5 k Cell viability of PC12 cells treated with various concentrations of Ang-NCs.

Table R2. The comparison of in vitro concentration and in vivo dosage among various works.

Num	Concentration	Dosage (mg/kg)	References
1	8 μ M	20	Nat. Commun. 2021, 12, 3393
2	100 μ M	10	ACS Nano 2023, 17, 16840
3	200 μ M	10	Adv. Mater. 2021, 34, 2106082
4	10 μ g/mL	10	ACS Nano 2023, 17, 5, 5033
5	10 μ g/mL	10	Nat. Commun. 2022,13,6835
6	100 μ g/mL	10	Adv. Sci. 2023, 10, 2206333
7	5 μ M	10	Cell Rep. Med. 2023 4, 101019
8	640 nM	10	Cancer Lett. 2010, 288, 251
9	30 μ M	10	Oncotarget. 2017, 8, 43237
10	100 μ g/mL	10	Our work

- 5) What is the mechanism of amyloid plaque reduction after the treatment using the probe. Is it involves macrophage-mediated clearance. Fig 6p Iba1 fluorescence staining results, only showed a reduction in the levels of Iba1+ microglia in the hippocampus. Please include further staining to clarify this. Please include staining on cortical region as well in the results.

Response: Thanks for your kind suggestion. According to the reports (*Chem. Bio. Chem.* 2021, 22, 2306; *Nanoscale* 2018, 10, 8989), the driving forces in aggregated and assembly process of A β monomer into fibril mainly were weak non-covalent bond forces, including hydrogen bonding and hydrophobic interaction among A β -A β monomers. The weak forces offer a chance for therapy probes with more stronger binding capability with A β monomers to disassemble the A β fibrils via competing with the self-aggregation interaction between A β monomers. Therefore, we customized the PTB molecule to compete and disrupt the aggregated force in fibril to achieve the reduction of amyloid plaque via the above interactions. Iba-1, a 17 kDa calcium-binding protein specifically expressed in macrophages/microglia (*Acta Cir Bras.* 2020,35, e202000406; *J. Cell Sci.* 2000, 113, 3073–3084), as a marker to microglia and macrophages (*Theranostics* 2022, 12, 5364; *Front. Cell. Neurosci.* 2022, 16, 921916) in the central nervous system. Furthermore, we increase the detailed observation of Iba-1 and GFAP in brain slices in the hippocampus and cortical regions. The fluorescence intensity of Iba1 and GFAP in two regions of Ang-NCs group greatly decreased to the level closed to that of WT group, indicating that the degraded product would not cause the significant activation of macrophages and microglia cells (**Fig.6i-k**). The results and explanation were added to the text.

Fig. 6 i Immunofluorescence images of A β plaques, Iba-1, and GFAP in hippocampus and cortex regions of ex vivo brain slices, red (A β plaque, Iba-1, and GFAP) and cell nucleus (blue), all scale bars: 100 μ m. **j, k** The corresponding fluorescence quantification of Iba-1 and GFAP.

- 6) The author stated that it is microscopic observation of A β fibril on ex vivo brain slices (Fig. 6h). Which antibody or dye was used. Is it fibril conformation specific? What about small forms of A β aggregates such as A β oligomers which are known to be more toxic than A β fibril. Please include further staining data and/or biochemical data to clarify this.

Response: Thanks for your kind suggestion. The A β fibril in **Fig. 6i** (original **Fig. 6h**) is neither conformation specific nor oligomer due to the four reasons: first, we increased further staining data respectively used the gold standard dye (ThS) and rabbit pAb anti-A β (Servicebio, GB111197, 1: 400 dilution), both were two fibril-specific probes but unresponsive to monomer and oligomer. The data revealed that massive A β plaques in APP/PS1 mice (AD) but almost was barely observable in WT and AD/Ang-NCs groups (**Supplementary Fig. 28**). Secondly, the red fluorescence outlined the size (50-70 μ m) of the detected species, which greatly exceed the normal size (10-20 nm) of oligomer (**Fig. 3f**), ruling out the possibility of oligomer again. Thirdly, the oligomer is unstable intermediate state from the monomer to fibril, which unable exist stably for long time and easily changed to fibril in physiological state. The brain slices in 7-month AD mice have possessed obvious severe neurological dysfunction and dyskinesia induced by massive A β plaques, which was far later than the time of the oligomer appearing. Lastly, we added the cell toxicity assay in PC12 and HT-22 cells to rule out the above speculation that the degraded A β fragments were toxic A β oligomers. The cell viability of PC12 and HT-22 cells treated with oligomer were 38.6% and 34.2%, that in the fibril group were 54.5% and 48.7%, while 93.4% and 99.6% in the A β fragment group (**Fig. 5m**), indicating negligible toxicity and certifying our speculation that the A β fragment is a safe short aggregate, not toxic oligomer. The detailed procedure and antibody or dye information were added to the text and the "Methods" section and show as follows:

A β plaque detection of ex vivo brain slices staining by ThS

To investigate the level of A β plaque in the brain of APP/PS1 mice after the treatment, the ex

vivo brains were first prepared into frozen slices. Then the slices were divided into two groups, one was immersed in the EDTA antigen retrieval buffer (pH = 8.0) and sent to the microwave for 8 min. After washing three times with phosphate-buffered saline (PBS, pH = 7.4), the nonspecific binding was blocked for 1 h by the 5% BSA and further incubated with the diluted primary antibody in PBS overnight at 4°C. Lastly, the slices were washed with PBS to remove the primary antibody and incubated with the secondary antibody for 1 h at room temperature. Another new slice was stained with ThS dye for confocal observation. Briefly, the slices were stained with ThS solution (100 μM) in 50% ethanol for 20 min and washed with 80% ethanol for 10 s and water for 10 s. After sealing with an anti-fluorescence quenching agent, these slices were observed by a confocal laser scanning microscope (Zeiss 880).

Fig. 6 i Immunofluorescence images of Aβ plaques in hippocampus and cortex regions of ex vivo brain slices, red (Aβ plaque) and cell nucleus (blue), all scale bars: 100 μm.

Supplementary Fig. 28 Microscopic observation of Aβ plaque (green spots) in hippocampus and cortex regions of ex vivo brain slices in WT mice, APP/PS1 (AD) mice, and AD mice with Ang-NCs. The green fluorescence was generated from the ThS dye.

Fig. 3 f TEM images of the morphology of Aβ oligomer.

Fig. 5 m Cell viability of PC12 and HT-22 cells incubated with A β oligomer (10 μ M), A β fibril (10 μ M), and degraded A β fragment (10 μ M).

- 7) Method details on brain extraction, brain slices, immunofluorescence staining and antibody information (for A β , GFAP, and iba1 staining) is missing.

Response: Thanks for your kind suggestion. The detailed preparation methods and procedures including brain extraction (collection), brain slices, immunofluorescence staining and antibody information (for A β , GFAP, and Iba-1 staining) were carefully added to text that were shown as follows:

Preparation and immunofluorescence staining of brain slices

The mice were anesthetized and perfused transcardially with saline after treatment. Then the brains of mice were collected and fixed via immersion in 4% paraformaldehyde for three days. After the dehydration treatment by gradient concentrations of alcohol, the brains were embedded into paraffin. The brain slices with 4 μ m of thickness are prepared with a pathology slicer (Leica RM2016) and mounted onto glass slides. The slices were deparaffinized after the successive treatment of xylene, alcohol, and washing with deionized water. Then these slices were immersed in the EDTA antigen retrieval buffer (pH = 8.0) and sent to the microwave for 8 min. After washed three times with phosphate-buffered saline (PBS) (pH = 7.4), the nonspecific binding was blocked for 1 h by bovine serum albumin (BSA, 5%) and further incubated with the diluted primary antibody in PBS overnight at 4°C. Lastly, the slices were washed to remove the primary antibody and incubated with the secondary antibody for 1 h at room temperature. Prior to the imaging observation using CaseViewer (Pannoramic MIDI), the slices were washed with PBS and stained with 4',6-diamidino-2-phenylindole (DAPI) (Servicebio G1012). The information of primary and secondary antibodies including Rabbit pAb anti-A β (Servicebio, GB111197, 1: 400 dilution), GFAP (Servicebio, GB11096, 1: 500 dilution), Iba-1 (Servicebio, GB113502, 1: 500 dilution), 8-OHdG antibody (Santa Cruz, sc-393871, 1: 50 dilution), Goat anti-rabbit IgG (Servicebio, GB21303, 1: 300 dilution), and Goat anti-mouse IgG Alexa Flour 488 (Abcom, ab150117, 1: 500 dilution).

- 8) Is there any consequences such as microbleeding that commonly observed in amyloid plaque clearance studies. Please provide histology or MRI evidence on this aspect.

Response: Thanks for your kind suggestion. The adverse risk of microbleeding was further assessed by the detailed histology observation. The brain of mice in various groups were harvested after treatment and prepared the slices with 4 μ m of thickness following the above procedure. The prussian was used as the dye to assess the level of microbleeds. The H&E images revealed both AD mice with Ang-NCs treatment and WT mice have no obvious iron-

positive deposits (**Supplementary Fig. 29**). The result powerfully demonstrated the high safety without microbleeds of our Ang-NCs in AD therapy process. The detailed experiment procedure was added in “Methods” section as follows:

Microbleeds observation

The microbleeds risk was investigated by the Prussian Blue histochemistry of brain slices. The slices were prepared according to the above procedure. Then the slices were treated with 2% potassium ferrocyanide in 2% HCl in Coplin jars for 1 h at room temperature, followed by a counterstain in the Nuclear fast Red solution (0.1%) for 5 min. Subsequently, the blue haemosiderin-deposition profiles in the stained brain slices were imaged by Grundium OCUS40X.

Supplementary Fig. 29 Microhemorrhage profile images of brain slices of AD mice treated with Ang-NCs and WT mice, both were stained with Prussian Blue Iron Stain Kit.

- 9) Toxicity test, It is stated that the healthy Balb/c mice were injected with 200 μ L of Ang-NCs (dosage: 10 mg/kg) and PBS. The whole blood was collected from the orbital of the mice on day 10 post the injection. However given that the treatment is 6 time injection over 2 weeks in APP/PS1 mice, a similar design is needed in the tox study to reflect the potential influence. Please include further supporting data on this.

Response: Thanks for your kind suggestion. For a more rigorous assessment of in vivo safety of our NCs in mice, the new toxicity test was repeated in the healthy Balb/c mice via harvesting the blood samples from the orbital of the mice on the eighteenth day post i.v. injection of Ang-NCs (dosage: 10 mg/kg) and PBS 6 times. The blood routine examination and blood biochemistry results were shown in **Supplementary Fig. 31** and indicated no significant difference between PBS and Ang-NCs groups, implying the Ang-NCs has a good biosafety. The result was shown as follows:

Supplementary Fig. 31 Blood routine examination and blood biochemistry. Blood were harvested from the healthy mice on the eighteenth day post i.v. injection of Ang-NCs (dosage: 10 mg/kg) and PBS six times (n = 3).

Reviewer #3:

The authors described a novel nanotheranostic system that targets the brain, with fibril-degrading and ROS-scavenging effects. They showed that the system was BBB permeable and was neuroprotective in vitro and in vivo. In vitro, the system protected cultured PC12 and HT-22 cells against A β fibrils. In AD model mice, the system not only reduced A β load, neuroinflammation and oxidative stress, but also improved functional outcome.

1) The authors claimed this was nanotheranostic system. While they showed its therapeutic application, they did not demonstrate any usefulness in imaging or diagnostic applications. As such, the rationale for designing a NIR-II probe was not justified.

Response: Thanks for your kind suggestion. The higher significance expected by the reviewer is consistent with the original intention of our design. Most previous studies have been limited to in vitro applications due to a lack of effective in vivo monitoring technology. The huge challenge is what we're trying to solve in this work. Therefore we must emphasize the multiple necessities to introduce the NIR imaging capabilities: 1) First, the common AD drugs have no imaging function and are unable to support the research on the interaction of drugs with A β fibril at the cellular level, which causes blind in vitro research for drug development. Although assisted by the A β -targeted commercial ThT probe, the introduction of ThT disturbs the interaction between the aimed probe and A β fibril, which cannot reflect the real interaction state. In our system, we performed massive in vitro tests including a binding kinetic test (**Fig. 3k-m**), inhibition kinetic test (**Supplementary Fig. 21**), specificity assessment (**Fig. 3n**), and in vitro blood brain barrier (BBB) penetration (**Fig.5c**) without the assistance of ThT, which demonstrated the necessary, usefulness, and correctness of our NIR imaging function. 2) Secondly, the brain has a unique BBB structure, which shields almost all drugs into the brain. The clinic positron emission tomography (PET) imaging probe (PiB) for AD diagnosis existing harmful radioactivity. Although many new nonradioactive fluorescent probes were developed to detect A β fibril in the brain. However, all probes faced low emission wavelengths below 725 nm. The low sensitivity visible imaging was unable to accurately monitor the microscopic events, e.g., real-time BBB penetration, targeted binding of probes with A β plaques in the brain, in vivo metabolism, and in vivo biodistribution. Differently, the long-wavelength NIR imaging at 1350 nm achieved these requirements to monitor the in vivo BBB penetration (**Fig. 6c**), accurately quantify the biodistribution with high accumulation in the spleen, not the kidney (false signal quantified by visible absorption) due to eliminating the self-absorption background (**Fig. 6d, e**). These demonstrate the high fidelity of our NIR imaging function that cannot achieved by visible imaging (**Fig. R6**). 3) The visual drugs not only provide direct evidence for the detailed in vivo study of the drug, the BBB-traversing process of entering the central brain system, and the binding with the target, but also can provide definite guiding principles to synthesize more aimed drug molecules. This powerful NIR tool breaks the traditional blind and ambiguous research model of tracer-free drugs, its intrinsic imaging function got rid of the limitations of the assistance of the ThT probe. The above massive important and necessary in vitro and in vivo exploration experiments would not be known if they lack NIR imaging function. We believe the "what you see is what you get" merit will stimulate new research growing points. We believe and hope more research about the unknown microscopic world would benefit from NIR imaging. Some of the above explanation was added to the text.

Fig.3 **k** Positive (blue) and negative (red) electrostatic potential analysis of the DFT-optimized PTB. **l** NIR imaging at 1350 nm of various concentrations of A β fibril with PTB molecule (2 μ M). **m** Binding kinetic process monitoring of A β fibrils (10 μ M) with various concentrations of PTB by fluorescence intensity variation at 1060 nm. **n** Absorption at 760 nm of various proteins, bovine serum albumin (BSA), and human serum albumin (HSA) treated with PTB.

Supplementary Fig. 21 **a** Relative fluorescence intensity of ThT (5 μ M) in the mixture of A β monomer (10 μ M) with various concentrations of PTB/TPTB-Ce NCs. **b** Inhibition kinetic curve and IC₅₀ value for PTB/TPTB-Ce NCs against A β monomers.

Fig.5 **c** Confocal images of *in vitro* BBB-penetration of two NCs by PC12 and HT-22 cells via Transwell model.

Fig. 6 c NIR imaging at 1350 nm of the brain in *APP/PS1* mice at various time points (0 and 12 h) post *i.v.* injection of two NCs (dosage: 10 mg/kg).

Fig. 6 d,e NIR images and corresponding fluorescence quantification for main organs of mice treated with NCs and Ang-NCs ($n = 4$), heart (H), liver (Li), spleen (S), lung (L), and kidney (K).

Fig. R6 a Optical density at 720 nm of various organs in *APP/PS1* mice injected with NCs and Ang-NCs (dosage: 10 mg/kg), and healthy mice treated with PBS ($n = 6$). **b** NIR image at 1350 nm of various organs from the healthy mice treated with Ang-NCs (dosage: 10 mg/kg), NCs (dosage: 10 mg/kg), and PBS.

2) For the therapeutic application, they provided a lot of data from *in vitro* and *in vivo*. But experimental details are generally lacking, making it difficult to interpret the quality of data. For example, they used *APP/PS1* transgenic AD mice in the *in vivo* experiments. The mice were 7-month-old when they were obtained, but it is not clear when the treatment started, and when the behavioral tests were performed. In addition, $n=6$ mice/group is a little too small for behavioral tests).

Response: Thanks for your kind suggestion. We not only enriched the experimental details and explanation of all original *in vitro* and *in vivo* assay but also added some new detailed procedures in the Methods section that were marked red in text (listed below). Moreover, to make clear the starting time of the treatment and the behavioral tests, we re-labeled the starting and ending time

of every process. The 7-month-old APP/PS1 mice were purchased and stayed for one day to adapt to the environment in advance (set day 0). The treatment started with the first injection on day 1, then the following successive five injections were performed on days 4, 7, 10, 13, and 16. After i.v. injections six times, the nesting assay was performed from day 18 to day 19. Subsequently, the Morris water maze (MWM) and novel object recognition (NOR) assays were performed from 20 to 25 and 26 to 28 days. After finishing the behavior tests, the brain and major organs of mice were harvested for slice observation from 29 to 35 days (**Fig. 6a**). The APP/PS1 mice treated with Ang-NCs showed improved recovery of memory loss, cognitive impairments, and behavioral impairments. Compared to the typical unimpaired paper pieces and the lowest behavior score in the AD group, the Ang-NCs group displayed a better performance and higher behavior score than that of the AD group (**Fig. 6l, m**). Subsequently, the long-term learning and memory capability of the spatial orientation of various groups were investigated by the Morris water maze (MWM) test. Under training with a platform for 5 days and testing after removing the platform, the time spent on moving from the start site to the target platform was recorded. The Ang-NCs group showed a reduced time and purposeful route to the platform (**Fig. 6n, o**). After removing platform, the staying time in target quadrant and crossing number also were higher than that of the AD group (**Fig. 6p, q**). At last, the novel object recognition (NOR) test was further administered. The target group demonstrated a higher recognition index (RI) compared to that of AD mice (**Fig. 6r**), which was comparable to that of the WT mice, reflecting great interest in exploring novel objects. The new results were shown as follows:

Fig. 6. a The outline of the experiment design and sequences for animal behavior evaluation.

Fig. 6. l, m The representative images and corresponding scoring comparisons of various groups in the nesting test ($n = 8$). **n** Swimming tracks of various groups. **o-q** were respectively the escape latency (**o**), the staying time in the target quadrant (**p**), and the crossing number through the platform (**q**) in the Morris water maze (MWM) test ($n = 8$).

Degradation of A β fibril and restructuring of fragment

To verify the disassembly ability of the NCs against A β fibril, 1 mL of PBS solution containing fibril (25 μ M) was co-incubated with NCs (200 μ g/mL) for 48 h at 37°C. Then the solution was

separated and the A β products were collected by centrifugation at 12,000 g for 20 min after adding 10 μ L of dimethyl sulfoxide (DMSO). The precipitate (fragment) was divided into two samples, one was supplemented with fresh A β monomer (final concentration: 25 μ M) for the restructuring growth. Another was added the same amount of A β monomer containing NCs (final concentration: 200 μ g/mL). Two samples were allowed to be co-incubated and grown for various times. Lastly, all samples were sent for TEM observation after staining with uranium acetate solution (1%).

Assessment of antioxidant, anti-inflammation effect, and A β ₁₋₄₂

To detect the antioxidant therapy effect of Ang-NCs, the brains of the APP/PS1 mice were harvested after i.v. injection of Ang-NCs one time. The DCFH-DA probe was dropped onto the surface of ex vivo brains. The fluorescence (excitation: 488 nm, emission: 525 nm) of brains was observed using an imaging system (IVIS Lumina III) post dropping. To investigate the levels of inflammation and A β ₁₋₄₂ in the brain, the brain of APP/PS1 (AD) mice was collected and homogenized in 0.2 mL of RIPA buffer containing protease and phosphatase inhibitors for 4 min (70 Hz) after i.v. injection of Ang-NCs six times. Then the solution was centrifuged at 100,000 g for 30 min and the supernatants were collected to detect the inflammatory mediators, i.e., IL-1 β and TNF- α by the corresponding ELISA kits. The pellet was further sonicated in 70% (w/v) formic acid followed by centrifugation at 100,000 g for 30 min at 4 °C again, and the supernatant containing A β ₁₋₄₂ was detected by ELISA kit. The concentration of proteins was quantified by a BCA kit.

Preparation and immunofluorescence staining of brain slices

The mice were anesthetized and perfused transcardially with saline after treatment. Then the brains of mice were collected and fixed via immersion in 4% paraformaldehyde for three days. After the dehydration treatment by gradient concentrations of alcohol, the brains were embedded into paraffin. The brain slices with 4 μ m of thickness are prepared with a pathology slicer (Leica RM2016) and mounted onto glass slides. The slices were deparaffinized after the successive treatment of xylene, alcohol, and washing with deionized water. Then these slices were immersed in the EDTA antigen retrieval buffer (pH = 8.0) and sent to the microwave for 8 min. After washed for three times with phosphate-buffered saline (PBS) (pH = 7.4), the nonspecific binding was blocked for 1 h by bovine serum albumin (BSA, 5%) and further incubated with the diluted primary antibody in PBS overnight at 4°C. Lastly, the slices were washed to remove the primary antibody and incubated with the secondary antibody for 1 h at room temperature. Prior to the imaging observation using CaseViewer (Pannoramic MIDI), the slices were washed with PBS and stained with 4',6-diamidino-2-phenylindole (DAPI) (Servicebio G1012). The information of primary and secondary antibodies including Rabbit pAb anti-A β (Servicebio, GB111197, 1: 400 dilution), GFAP (Servicebio, GB11096, 1: 500 dilution), Iba-1 (Servicebio, GB113502, 1: 500 dilution), 8-OHdG antibody (Santa Cruz, sc-393871, 1: 50 dilution), Goat anti-rabbit IgG (Servicebio, GB21303, 1: 300 dilution), and Goat anti-mouse IgG Alexa Flour 488 (Abcom, ab150117, 1: 500 dilution).

Microbleeds observation

The microbleeds risk was investigated by the Prussian Blue histochemistry of brain slices. The slices were prepared according to the above procedure. Then the slices were treated with 2% potassium ferrocyanide in 2% HCl in Coplin jars for 1 h at room temperature, followed by a

counterstain in the Nuclear fast Red solution (0.1%) for 5 min. Subsequently, the blue haemosiderin-deposition profiles in the stained brain slices were imaged by Grundium OCUS40X.

A β plaque detection of ex vivo brain slices staining by ThS

To investigate the level of A β plaque in the brain of APP/PS1 mice after the treatment, the ex vivo brain slices were respectively stained by Rabbit pAb anti-A β probe and ThS dye. First, the slice was immersed in the EDTA antigen retrieval buffer (pH = 8.0) and sent to the microwave for 8 min. After washing for three times with phosphate-buffered saline (PBS, pH = 7.4), the nonspecific binding was blocked for 1 h by the 5% BSA and further incubated with the diluted primary antibody in PBS overnight at 4°C. The slices were washed with PBS to remove the primary antibody and incubated with the secondary antibody for 1 h at room temperature. Secondly, for the ThS staining, the new slice was stained with ThS dye for confocal observation. Briefly, the slices were stained with ThS solution (100 μ M) in 50% ethanol for 20 min and washed with 80% ethanol for 10 s and water for 10 s. After sealing with an anti-fluorescence quenching agent, these slices were observed by a confocal laser scanning microscope (Zeiss 880).

- 3) Another major concern is lack of control. Most of the in vitro and in vivo experiments were conducted without proper control.

Response: Thanks for your kind suggestion. We respectively increase many in vitro controls including the comparison of antioxidant, A β decomposition, and toxicity reverse assays of the fibril between sole PTB and sole TPTB-Ce. Moreover, we added the control assay to explore the toxicity of oligomer and degraded fragments to PC12 and HT-22 cells. First, the laser confocal microscopy result indicated strong green fluorescence of A β fibrils appeared in the TPTB-Ce with A β fibrils group and A β fibrils group, while it faded in the PTB with A β fibrils group (**Supplementary Fig. 25a**), suggesting the effective disassembly capability of PTB to A β fibrils than TPTB-Ce. Subsequently, we further detected the antioxidant ability of two molecules using a reactive oxygen species (ROS) probe (DCFH-DA). The quantification of the oxidative product (DCF) indicated the strong antioxidant ability of TPTB-Ce, which greatly exceeds PTB (**Supplementary Fig. 25b**). Furthermore, the therapy output of reverse neurotoxicity was measured by the CCK-8 assay. The results showed both two molecules revealed superior neuroprotective function (**Fig. 5l**). Lastly, the toxicity of A β oligomer, fibrils, and the fragment were tested. The cell viability of PC12 and HT-22 cells treated with oligomer were 38.6% and 34.2%, that in the fibril group were 54.5% and 48.7%, while 93.4% and 99.6% in the A β fragment group (**Fig. 5m**). The results suggest that the Ang-NCs therapy was safe and the A β fragment is a safe species, not a toxic species. Because the focus of in vitro and in vivo research was different, the therapy contribution of PTB and TPTB-Ce was defined by in vitro assays. The main focus of in vivo exploration was on whether the Ang-NCs could penetrate the BBB and trigger the following therapy effect to achieve behavioral and cognitive improvements in the AD mice model. Therefore, we increase a series of detailed investigations of in vivo therapy functions (**Fig. 6g-k**) by quantification of fluorescence intensity and ELISA assays besides increasing the sample number of all in vivo biodistribution (**Fig. 6d, e**) and behavioral experiments (**Fig. 6m-r**). The results indicated improved behavior and memory in AD mice after Ang-NCs treatment.

Supplementary Fig. 25 a Confocal images of PC12 cells incubated with PBS, A β fibril, A β fibril with TPTB-Ce, and A β fibril with PTB for 48 h. All cells were stained with ThT (green fluorescence). **b** The ROS level of three samples tested by the microplate reader and stained with DCFH-DA probe. The concentrations of two AIEgens and A β fibril were respectively 100 μ g/mL and 10 μ M.

Fig. 5 l Cell viability of PC12 cells treated with A β fibril, A β fibril with PTB (100 μ g/mL), and A β fibril with TPTB-Ce (100 μ g/mL) for 48 h. The concentrations of A β fibril were 10 μ M. **m** Cell viability of PC12 and HT-22 cells incubated with A β oligomer (10 μ M), A β fibril (10 μ M), and degraded A β fragment (10 μ M).

Fig. 6 g The levels of TNF- α , IL-1 β inflammatory factors, and A β_{1-42} in the various mice. **h** The fluorescence quantification of A β

plaques in panel i. i Immunofluorescence images of A β plaques, Iba-1, and GFAP in hippocampus and cortex regions of ex vivo brain slices, red (A β plaque, Iba-1, and GFAP) and cell nucleus (blue), all scale bars: 100 μ m. j, k The corresponding fluorescence quantification of Iba-1 and GFAP.

Fig. 6 d, e NIR images and corresponding fluorescence quantification for main organs of mice treated with NCs and Ang-NCs (n = 4), heart (H), liver (Li), spleen (S), lung (L), and kidney (K).

Fig. 6 m The corresponding scoring comparisons of various groups in the nesting test (n = 8). **o-q** were respectively the escape latency (**o**), the time in the target quadrant (**p**), and the crossing number through the platform (**q**) in the Morris water maze (MWM) test (n = 8). **r** The recognition index in the NOR test (n = 8).

4) They showed A β fibril staining after treatment (Fig 6h), however, no quantitative data were provided. Similarly, no quantitative data for the immunofluorescence of Iba1 and GFAP. In addition, to show the A β fibril-reducing effects, the authors need to provide additional evidence, with another measurement of A β , for example, western or ELISA.

Response: Thanks for your kind suggestion. The quantitative data of A β fibril staining after treatment was added in **Fig. 6h, i**. The result revealed effective degradation of A β plaque by Ang-NCs. Furthermore, we used the ELISA measurement to further test the level of A β_{1-42} in the various mice (**Fig. 6g**). Briefly, the homogenate of the ex vivo brains in APP/PS1 mice were co-incubated with a specific detection reagent for A β fibril and detected the OD at 450 nm by a microplate reader. As expected, the AD group has high level of A β fibril, while the treatment group indicated almost same decreased level of fibril with that in PBS group, both powerfully verified the degradation function of our Ang-NCs to A β fibril. Moreover, we explored the immunofluorescence images (**Fig. 6i**) and fluorescence quantification of Iba-1 (**Fig. 6j**) and

GFAP (**Fig. 6k**) in the hippocampal and cortex regions of brain slices in various mice. The decreased level of Iba-1 and GFAP in AD demonstrate the Ang-NCs treatment greatly normalized these levels close to that in WT mice, suggesting no obvious activation of astrocytes and microglia cells.

Fig. 6 h The fluorescence quantification of Aβ plaques in panel i. **i** Immunofluorescence images of Aβ plaques in hippocampus and cortex regions of ex vivo brain slices, red (Aβ plaque) and cell nucleus (blue), all scale bars: 100 μm.

Fig. 6 g The levels of Aβ₁₋₄₂ in the various mice.

Fig. 6 i Immunofluorescence images of Aβ plaques, Iba-1, and GFAP in hippocampus and cortex regions of ex vivo brain slices, red (Aβ plaque, Iba-1, and GFAP) and cell nucleus (blue), all scale bars: 100 μm. **j**, **k** The corresponding fluorescence quantification of Iba-1 and GFAP.

5) Similarly, the evidence of in vivo anti-oxidative stress is rather weak (Fig 6f). It was from ex vivo brain imaging (again no mention of how this was done in the “Methods” sections). There are

better ways to assess oxidative damage.

Response: Thanks for your kind suggestion. Moreover, to strengthened the evidence of in vivo anti-oxidative stress, the detailed immunofluorescence staining imaging of brain slices were further assessed. The 8-hydroxy-2'-deoxyguanosine (8-OHdG), as an oxidative stress hallmark, was utilized to determine the level of ROS in the brain (*Antioxidants* 2020, 9, 1209-1224; *Adv. Mater.* 2021, 33, 2100746). The reduced green fluorescence result revealed that the Ang-NCs could efficiently decrease the oxidation level in the brain of APP/PS1 mice (**Supplementary Fig. 27**). The detailed procedure for the anti-oxidative stress of ex vivo brain imaging was added in "Methods" sections and were listed as follows:

Preparation and immunofluorescence staining of brain slices

The mice were anesthetized and perfused transcardially with saline after treatment. Then the brains of mice were collected and fixed via immersion in 4% paraformaldehyde for three days. After the dehydration treatment by gradient concentrations of alcohol, the brains were embedded into paraffin. The brain slices with 4 μm of thickness are prepared with a pathology slicer (Leica RM2016) and mounted onto the glass slides. The slices were deparaffinized after the successive treatment of xylene, alcohol, and washing with deionized water. Then these slices were immersed in the EDTA antigen retrieval buffer (pH = 8.0) and sent to the microwave for 8 min. After washed three times with phosphate-buffered saline (PBS) (pH = 7.4), the nonspecific binding was blocked for 1 h by bovine serum albumin (BSA, 5%) and further incubated with the diluted primary antibody in PBS overnight at 4°C. Lastly, the slices were washed to remove the primary antibody and incubated with the secondary antibody for 1 h at room temperature. Prior to the imaging observation using CaseViewer (Pannoramic MIDI), the slices were washed with PBS and stained with 4',6-diamidino-2-phenylindole (DAPI) (Servicebio G1012). The information of primary and secondary antibodies including 8-OHdG antibody (Santa Cruz, sc-393871, 1: 50 dilution) and Goat anti-mouse IgG Alexa Flour 488 (Abcom, ab150117, 1: 500 dilution).

Assessment of antioxidant, anti-inflammation effect, and A β ₁₋₄₂

To detect the antioxidant therapy effect of Ang-NCs, the brains of the APP/PS1 mice were harvested after i.v. injection of Ang-NCs one time. The DCFH-DA probe was dropped onto the surface of ex vivo brains. The fluorescence (excitation: 488 nm, emission: 525 nm) of brains was observed using an imaging system (IVIS Lumia III) post dropping. To investigate the levels of inflammation and A β ₁₋₄₂ in the brain, the brain of APP/PS1 (AD) mice was collected and homogenized in 0.2 mL of RIPA buffer containing protease and phosphatase inhibitors for 4 min (70 Hz) after i.v. injection of Ang-NCs six times. Then the solution was centrifuged at 100,000 g for 30 min and the supernatants were collected to detect the inflammatory mediators, i.e., IL-1 β and TNF- α by the corresponding ELISA kits. The pellet was further sonicated in 70% (w/v) formic acid followed by centrifugation at 100,000 g for 30 min at 4 °C again, and the supernatant containing A β ₁₋₄₂ was detected by ELISA kit. The concentration of proteins was quantified by a BCA kit.

Supplementary Fig. 27 a Immunofluorescence images of 8-hydroxy-2'-deoxyguanosine (8-OHdG) in hippocampus and cortex regions of ex vivo brain slices from APP/PS1 mice (AD) and WT mice after treatment. **b** The corresponding fluorescence intensity quantification of 8-OHdG.

6) In Fig 6c,d, the scale bars seem incorrect (the dark end represents higher signal?). Also, the images from Fig. 6d don't support the observation of "a major accumulation in the liver and kidney". The strength of signals in the liver and kidney are completely different.

Response: Thanks for your kind suggestion. The color bar was incorrect and has been revised to the dark end represents lower signal. There is a difference between the NIR fluorescence intensity and the optical density (OD) quantification of homogenate, i.e., the high level of nanoparticles distributed in the liver and spleen in the former, and the liver, spleen, and kidney for the latter. We speculated it may be generated by two reasons: one reason was the difference of the test wavelengths and test mode, i.e., absorption at 720 nm and emission at 1350 nm. The other may be induced by the background value is not deducted, i.e., the OD values were the absolute OD without considering the self-absorption in the visible region of organs. Therefore, we simultaneously re-tested the OD at 720 nm of homogenate from healthy C57 mice and the APP/PS1 mice injected with NCs and Ang-NCs treatments (n = 6), whose result proves our hypothesis (**Fig. R6a**), revealing high self-absorption background at 720 nm tested by a microplate reader in healthy mice. The liver, spleen, and kidney of APP/PS1 mice have higher accumulations than other organs. Differently, the NIR imaging eliminates the above interference of the visible background of organs and could reflect the distribution level more accurately. As shown in **Fig. R6b**, the various organs of healthy mice treated with NCs and Ang-NCs revealed main accumulation in the liver and spleen, while the healthy mice injected with PBS have no signal, suggesting a high signal-to-noise ratio of NIR imaging. Therefore, to acquire a more accurate accumulation of Ang-NCs, the fluorescence intensity of NIR imaging was used to quantify the distribution of two NCs in different organs and increased the number of APP/PS1 mice from 1 to 4. The results indicated both two nanocomposites were mainly distributed in the liver and spleen (**Fig. 6d, e**). The similar biodistribution result was common in the previous works shown in **Fig. R7b** (*Nat. Commun.* 2021, 12, 3393).

Fig. R6 a Optical density at 720 nm of various organs in APP/PS1 mice injected with NCs and Ang-NCs (dosage: 10 mg/kg), and healthy mice treated with PBS (n = 6). **b** NIR image at 1350 nm of various organs from the healthy mice treated with Ang-NCs (dosage: 10 mg/kg), NCs (dosage: 10 mg/kg), and PBS.

Fig. 6 d, e NIR images and corresponding fluorescence quantification for main organs of mice treated with NCs and Ang-NCs (n = 4), heart (H), liver (Li), spleen (S), lung (L), and kidney (K).

Fig. R7 b Distribution of Si element in main organs (heart, liver, spleen, lung, and kidney) and tumors of mice at varied time points after FHPG administration.

7) Minor. The author seemed to use “fibrosis” to describe the formation of A β fibrils. This is not common in the AD field, since the term “fibrosis” usually refers to the accumulation of fibrous connective tissue.

Response: Thanks for your kind suggestion. All expressions of “fibrosis” were revised to “**fibril generation**” or “**formation of A β fibrils**” that were marked by red.

Reviewers' Comments:

Reviewer #1:

Remarks to the Author:

The comments have been addressed sufficiently and the manuscript is improved, suitable for publication in Nature Communications.

Reviewer #2:

Remarks to the Author:

The manuscript has been greatly improved. The authors have included solid additional data and have addressed almost all my questions.

Minor correction is needed:

According to the 'Sex and Gender Equity in Research – SAGER – guidelines' of the journal, the title and/or abstract must indicate when findings apply to only one sex or gender. In the current version of the manuscript, the sex of the mice has now been mentioned in the method section (3 times). Please add information on the sex of mice in the title, abstract and in the results when necessary. Please include a sentence on this issue in the limitation section of the study as well. This is especially relevant because there is a sex difference in the pathology level in mouse models of Alzheimer's disease. The treatment effect might differ between male and female mouse model of Alzheimer's disease.

Reviewer #3:

Remarks to the Author:

The authors have considered and addressed my comments. They have made appropriate revisions and provided necessary clarifications, and have incorporated my suggestions into the revised manuscript. No further questions.

List of changes: Point-to-Point Responses to the Comments of Reviewers

Reviewer: #2

Comments: The manuscript has been greatly improved. The authors have included solid additional data and have addressed almost all my questions.

Minor correction is needed:

According to the 'Sex and Gender Equity in Research – SAGER – guidelines' of the journal, the title and/or abstract must indicate when findings apply to only one sex or gender. In the current version of the manuscript, the sex of the mice has now been mentioned in the method section (3 times). Please add information on the sex of mice in the title, abstract and in the results when necessary. Please include a sentence on this issue in the limitation section of the study as well. This is especially relevant because there is a sex difference in the pathology level in mouse models of Alzheimer's disease. The treatment effect might differ between male and female mouse model of Alzheimer's disease.

Response: Thanks for your kind suggestion. The requirement also was made by the Author_Checklist_ from editorial office "To comply with SAGER guidelines, if the research findings apply to only one sex or gender this must be indicated in the title and abstract. For example, if only female mice were used for the in vivo experiments this should be specified in the Abstract (as well as in the Methods)". Therefore, we added the explanations of the sex (female) of the mice as many times as possible, in the abstract section, results (legend in Fig. 6 and text in Line 1 of Page 9), and Method sections (3 times). Moreover, we also added the sex illustration "This study addresses the drug against A β plaque, so the **female** APP/PS1 mice that has higher levels of A β burdens were selected for the above research" to the Reporting on sex item of Reporting Summary file. We believe that the reader has acquired the certain and sufficient sex information of mice from seven times of repeated explanations that were shown as follows:

Abstract

Another effectively scavenges ROS and inflammation to remodel the cerebral redox balance and enhance the therapy effect, together reversing the neurotoxicity and achieving effective behavioral and cognitive improvements in the **female** AD mice model.

Fig. 6 | In vivo theranostic of Ang-NCs in the **female** APP/PS1 (AD) mice.

Line 1 in Page 9 "To explore the in vivo dual-targeted therapy function of Ang-NCs in the **female** AD mice after six times of injections, we first analyzed in situ ROS change in the ex vivo brain after i.v. injection of Ang-NCs one time, the results revealed an efficient decrease of excess ROS induced by A β fibril and inflammation (Fig. 6f)".

Animals

All AD mice and littermate control in this work were seven-month-old **female** APP/PS1 transgenic AD mice that were purchased from Beijing HFK Bioscience Co. Ltd. (Beijing, China). The wild-type C57 mice and healthy Balb/c white mice were **female** and purchased from SiPeiFu (SPF) Biotechnology Co., Ltd. (Beijing, China).

In vivo NIR- II imaging

The seven-month-old **female** APP/PS1 mice were selected as AD mouse models. First, the Ang-NCs were i.v. injected into the AD mice (dosage: 10 mg/kg).